# Frequency Domain-based Dataset Distillation

**Donghyeok Shin**[*]
KAIST
tlsehdgur0@kaist.ac.kr

**Seungjae Shin**[*]
KAIST
tmdwo0910@kaist.ac.kr

**Il-Chul Moon**
KAIST, Summary.AI
icmoon@kaist.ac.kr

## Abstract

This paper presents FreD, a novel parameterization method for dataset distillation, which utilizes the frequency domain to distill a small-sized synthetic dataset from a large-sized original dataset. Unlike conventional approaches that focus on the spatial domain, FreD employs frequency-based transforms to optimize the frequency representations of each data instance. By leveraging the concentration of spatial domain information on specific frequency components, FreD intelligently selects a subset of frequency dimensions for optimization, leading to a significant reduction in the required budget for synthesizing an instance. Through the selection of frequency dimensions based on the explained variance, FreD demonstrates both theoretical and empirical evidence of its ability to operate efficiently within a limited budget, while better preserving the information of the original dataset compared to conventional parameterization methods. Furthermore, based on the orthogonal compatibility of FreD with existing methods, we confirm that FreD consistently improves the performances of existing distillation methods over the evaluation scenarios with different benchmark datasets. We release the code at https://github.com/sdh0818/FreD.

## 1 Introduction

The era of big data presents challenges in data processing, analysis, and storage; and researchers have studied the concept of *dataset distillation* [20, 26, 14] to resolve these challenges. Specifically, the objective of dataset distillation is to synthesize a dataset with a smaller cardinality that can preserve the performance of original large-sized datasets in machine learning tasks. By distilling the key features from the original dataset into a condensed dataset, less computational resources and storage space are required while maintaining the performances from the original dataset. Dataset distillation optimizes a small-sized variable to represent the input, not the model parameters. This optimization leads the variable to store a synthetic dataset, and the variable is defined in the memory space with the constraint of limited capacity. Since dataset distillation involves the optimization of *data variables* with limited capacity, distillation parameter designs, which we refer to as parameterization, could significantly improve corresponding optimization while minimizing memory usage.

Some of the existing distillation methods, i.e. 2D image dataset distillations [24, 14, 3, 25, 27], naively optimize data variables embedded on the input space without any transformation or encoding. We will refer to distillation on the provided input space as spatial domain distillation, as an opposite concept of frequency domain distillation that is the focus of this paper. The main drawback of spatial domain distillation would be the difficulty in specifying the importance of each pixel dimension on the spatial domain, so it is necessary to utilize the same budget as the original dimension for representing a single instance. From the perspective of a data variable, which needs to capture the key information of the original dataset with a limited budget, the variable modeling with whole dimensions becomes a significant bottleneck that limits the number of distilled data instances. Various spatial domain-based

---

[*] Equal contribution

37th Conference on Neural Information Processing Systems (NeurIPS 2023).

parameterization methods [7, 10] have been proposed to overcome this problem, but they are either highly vulnerable to instance-specific information loss [7] or require additional training with an auxiliary network [10].

This paper argues that the spatial domain distillation has limitations in terms of 1) memory efficiency and 2) representation constraints of the entire dataset. Accordingly, we propose a novel Frequency domain-based dataset Distillation method, coined FreD, which contributes to maintaining the task performances of the original dataset based on the limited budget. FreD employs a frequency-based transform to learn the data variable in the transformed frequency domain. Particularly, our proposed method selects and utilizes a subset of frequency dimensions that are crucial for the formation of an instance and the corresponding dataset. By doing so, we are able to achieve a better condensed representation of the original dataset with even fewer dimensions, which corresponds to significant efficiency in memory. Throughout the empirical evaluations with different benchmarks, FreD shows consistent improvements over the existing methods regardless of the distillation objective utilized.

## 2 Preliminary

### 2.1 Basic Notations

This paper primarily focuses on the dataset distillation for classification tasks, which is a widely studied scenario in the dataset distillation community [20, 26]. Given $C$ classes, let $\mathcal{X} \in \mathbb{R}^d$ denote the input variable space, and $\mathcal{Y} = \{1, 2, ..., C\}$ represent the set of candidate labels. Our dataset is $D = \{(x_i, y_i)\}_{i=1}^{N} \subseteq \mathcal{X} \times \mathcal{Y}$. We assume that each instance $(x, y)$ is drawn i.i.d from the data population distribution $\mathcal{P}$. Let a deep neural network $\phi_\theta : \mathcal{X} \to \mathcal{Y}$ be parameterized by $\theta \in \Theta$. This paper employs cross-entropy for the generic loss function, $\ell(x, y; \theta)$.

### 2.2 Previous Researches: Optimization and Parameterization of Dataset Distillation

**Dataset Distillation Formulation.** The goal of dataset distillation is to produce a cardinality-reduced dataset $S$ from the given dataset $D$, while maximally retaining the task-relevant information of $D$. The objective of dataset distillation is formulated as follows:

$$\min_{S} \mathbb{E}_{(x,y) \in \mathcal{P}}[\ell(x, y; \theta_S)] \text{ where } \theta_S = \arg\min_{\theta} \frac{1}{|S|} \sum_{(x_i, y_i) \in S} \ell(x_i, y_i; \theta) \tag{1}$$

However, the optimization of Eq. (1) is costly and not scalable since it is a bi-level problem [26, 17]. To avoid this problem, various proxy objectives are proposed to match the task-relevant information of $D$ such as gradient [26], features [25], trajectory [3], etc. Throughout this paper, we generally express these objectives as $\mathcal{L}_{DD}(S, D)$.

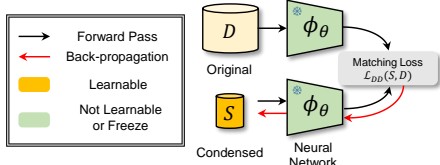

(a) Dataset distillation on input-sized variable.

**Parameterization of $S$.** Orthogonal to the investigation on objectives, other researchers have also examined the parameterization and corresponding dimensionality of the data variable, $S$. By parameterizing $S$ in a more efficient manner, rather than directly storing input-sized instances, it is possible to distill more instances and enhance the representation capability of $S$ for $D$ [7, 10]. Figure 1 divides the existing methods into three categories.

As a method of Figure 1b, HaBa [10] proposed a technique for dataset factorization, which involves breaking the dataset into bases and hallucination networks for diverse samples. However, incorporating an additional network in distillation requires a separate budget, which is distinct from the data instances.

As a method of Figure 1c, IDC [7] proposed the utilization of an upsampler module in dataset distillation

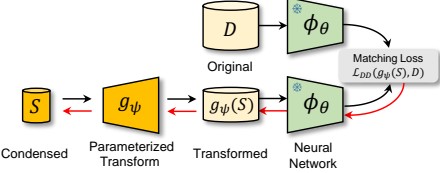

(b) Dataset distillation on variable with parameterized transform.

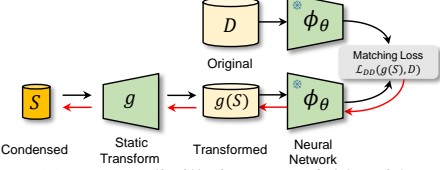

(c) Dataset distillation on variable with static transform.

Figure 1: Comparison of different parameterization strategies for optimization of $S$.

to increase the number of data instances. This parameterization enables the condensation of a single instance with reduced spatial dimensions, thereby increasing the available number of instances. However, the compression still operates on the spatial domain, which results in a significant information loss per instance. Please refer to Appendix A.1 for the detailed literature reviews.

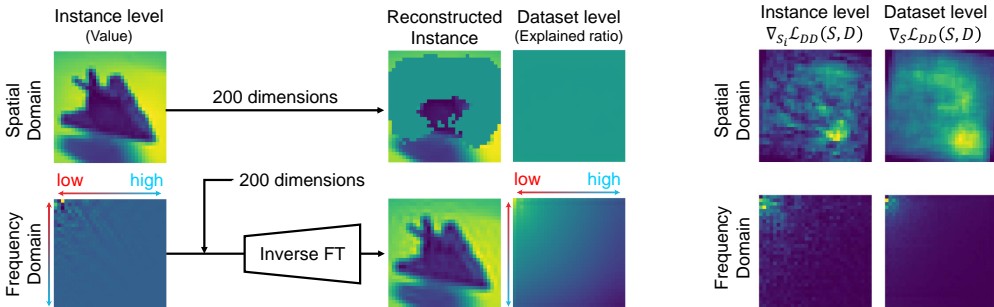

(a) Statistical properties of dataset on spatial and frequency domain.    (b) Tendency of dataset distillation loss.

Figure 2: Visualization of the information concentrated property. (a) On the frequency domain, a large proportion of both amplitude (left) and explained variance ratio (right) are concentrated in a few dimensions. (b) The magnitude of the gradient for $\mathcal{L}_{DD}(S, D)$ is also concentrated in a few frequency dimensions. We utilize trajectory matching for $\mathcal{L}_{DD}(S, D)$. A brighter color denotes a higher value. Best viewed in color.

## 3 Methodology

### 3.1 Motivation

The core idea of this paper is to utilize a frequency domain to compress information from the spatial domain into a small number of frequency dimensions. Therefore, we briefly introduce the frequency transform, which is a connector between spatial and frequency domains. Frequency-transform function, denoted as $\mathcal{F}$, converts an input signal $x$ to the frequency domain. It results in the frequency representation, $f = \mathcal{F}(x) \in \mathbb{R}^{d_1 \times d_2}$.[2] The element-wise expression of $f$ with $x$ is as follows:

$$f_{u,v} = \sum_{a=0}^{d_1-1} \sum_{b=0}^{d_2-1} x_{a,b}\, \phi(a, b, u, v) \tag{2}$$

Here, $\phi(a, b, u, v)$ is a basis function for the frequency domain, and the form of $\phi(a, b, u, v)$ differs by the choice of specific transform function, $\mathcal{F}$. In general, an inverse of $\mathcal{F}$, $\mathcal{F}^{-1}$, exists so that it enables the reconstruction of the original input $x$, from its frequency components $f$ i.e. $x = \mathcal{F}^{-1}(\mathcal{F}(x))$.

A characteristic of the frequency domain is that there exist specific frequency dimensions that encapsulate the major information of data instances from the other domain. According to [1, 21], natural images tend to exhibit the energy compaction property, which is a concentration of energy in the low-frequency region. Consequently, it becomes feasible to compress the provided instances by exclusively leveraging the low-frequency region. Remark 1 states that there exists a subset of frequency dimensions that minimize the reconstruction error of a given instance as follows:

**Remark 1.** *([16]) Given $d$-dimension data instance of a signal $x = [x_0, ..., x_d]^T$, let $f = [f_0, ..., f_d]^T$ be its frequency representation with discrete cosine transform (DCT), i.e. $f = DCT(x)$. Also, let $f^{(k)}$ denote the $k$ elements of $f$ while other elements are 0. Then, the minimizer of the reconstruction error $\|x - DCT^{-1}(f^{(k)})\|_2$ is $[f_0, , , , f_k, 0, ...0]$.*

Figure 2a supports the remark. When we apply a frequency transform to an image, we can see that the coefficients in the low-frequency region have very large values compared to the coefficients in other regions. Also, as shown in Remark 1, the reconstructed image is quite similar to the original image while only utilizing the lowest frequency dimensions. Beyond the instance level, The right side of Figure 2a represents that certain dimensions in the frequency domain exhibit a concentration of variance. It suggests that the frequency domain requires only a small number of dimensions to

---
[2]Although some frequency transform handles the complex space, we use the real space for brevity.

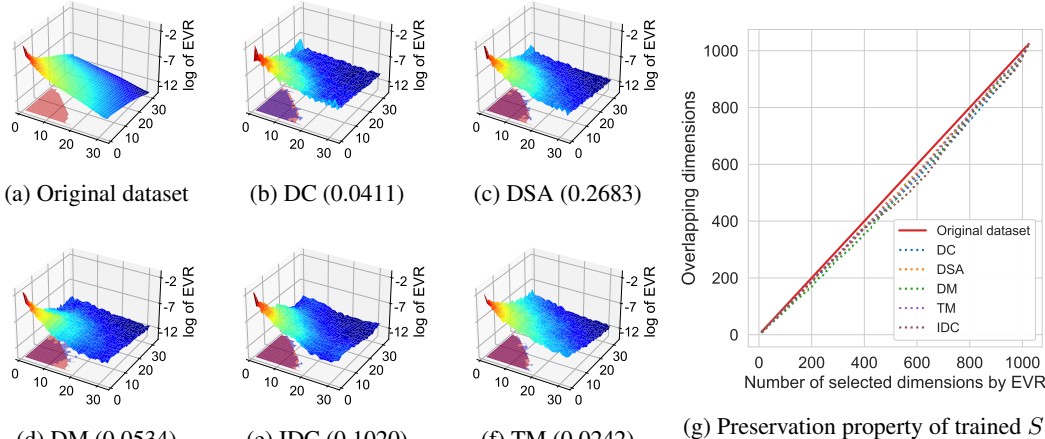

(a) Original dataset      (b) DC (0.0411)      (c) DSA (0.2683)

(d) DM (0.0534)      (e) IDC (0.1020)      (f) TM (0.0242)     (g) Preservation property of trained $S$

Figure 3: (a)-(f) The $\log$ of explained variance ratio on each frequency domain dimension (surface) and the degree of overlap between $D$ and $S$ for the top-200 dimensions (floor). The parenthesis is $L_2$ difference of explained variance ratio between $D$ and $S$. (g) It can be observed that trained $S$ mostly preserves the top-$k$ explained variance ratio dimensions of $D$, regardless of $k$.

describe the total variance of a dataset. On the other hand, the spatial domain does not exhibit this behavior; poor quality of reconstruction, and a similar variance ratio across all dimensions.

To investigate whether this tendency is maintained in the dataset distillation, we conducted a frequency analysis of various dataset distillation losses $\mathcal{L}_{DD}(S, D)$. Figure 2b shows the magnitude of gradient for $\mathcal{L}_{DD}(S, D)$ i.e. $\|\nabla_S \mathcal{L}_{DD}(S, D)\|$. In contrast to the spatial domain, where the gradients w.r.t the spatial domain are uniformly distributed across each dimension; the gradients w.r.t frequency domain are condensed into certain dimensions. This result suggests that many dimensions are needed in the spatial domain to reduce the loss, but only a few specific dimensions are required in the frequency domain. Furthermore, we compared the frequency domain information of $D$ and $S$ which were trained with losses from previous research. As shown in Figures 3a to 3f, the distribution of explained variance ratios in the frequency domain is very similar across different distillation losses. Also, Figure 3g shows that trained $S$ mostly preserves the top-$k$ explained variance ratio dimensions of $D$ in the frequency domain, regardless of $k$ and the distillation loss function. Consequently, if there exists a frequency dimension on $D$ with a low explained variance ratio of frequency representations, it can be speculated that the absence of the dimension will have little impact on the optimization of $S$.

## 3.2   FreD: Frequency domain-based Dataset Distillation

We introduce the frequency domain-based dataset distillation method, coined FreD, which only utilizes the subset of entire frequency dimensions. Utilizing FreD on the construction of $S$ has several advantages. First of all, since the frequency domain concentrates information in a few dimensions, FreD can be easy to specify some dimensions that preserve the most information. As a result, by reducing the necessary dimensions for each instance, the remaining budget can be utilized to increase the number of condensed images. Second, FreD can be orthogonally applied to existing spatial-based methods without constraining the available loss functions. Figure 4 illustrates the overview of FreD with the information flow. FreD consists of three main components: 1) *Synthetic frequency memory*, 2) *Binary mask memory*, and 3) *Inverse frequency transform*.

**Synthetic Frequency Memory** $F$**.**   Synthetic frequency memory $F$ consists of $|F|$ frequency-label data pair, i.e. $F = \{(f^{(i)}, y^{(i)})\}_{i=1}^{|F|}$. Each $f^{(i)} \in \mathbb{R}^d$ is initialized with a frequency representation acquired through the frequency transform of randomly sampled $(x^{(i)}, y^{(i)}) \sim D$, i.e. $f^{(i)} = \mathcal{F}(x^{(i)})$.

**Binary Mask Memory** $M$**.**   To filter out uninformative dimensions in the frequency domain, we introduce a set of binary masks. Assuming the disparity of each class in the frequency domain, we utilize class-wise masks as $M = \{M^{(1)}, ..., M^{(C)}\}$, where each $M^{(c)} \in \{0, 1\}^d$ denotes the binary mask of class $c$. To filter out superfluous dimensions, each $M^{(c)}$ is constrained to have $k$ non-zero elements, with the remaining dimensions masked as zero. Based on $M^{(c)}$, we acquire the class-wise

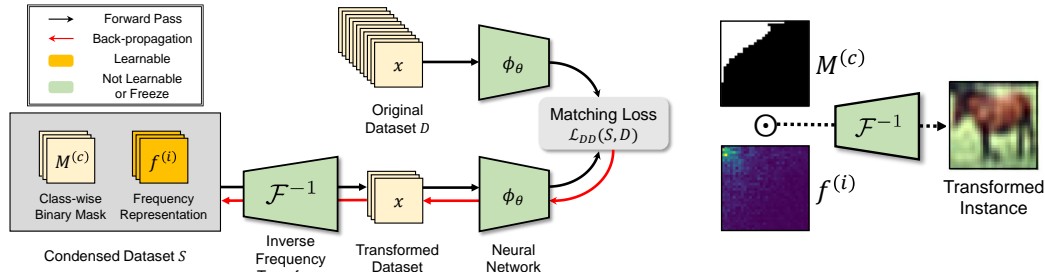

(a) Overall structure of FreD.

(b) Inverse frequency transform with binary masking.

Figure 4: Visualization of the proposed method, Frequency domain-based Dataset Distillation, FreD.

filtered frequency representation, $M^{(c)} \odot f^{(i)}$, which takes $k$ frequency coefficients as follows:

$$M^{(c)}_{u,v} \odot f^{(i)} = \begin{cases} f^{(i)}_{u,v} & \text{if } M^{(c)}_{u,v} = 1 \\ 0 & \text{otherwise} \end{cases} \tag{3}$$

PCA-based variance analysis is commonly employed for analyzing the distribution data instances. This paper measures the informativeness of each dimension in the frequency domain by utilizing the Explained Variance Ratio (EVR) of each dimension, which we denote as $\eta_{u,v} = \frac{\sigma^2_{u,v}}{\sum_{u',v'} \sigma^2_{u',v'}}$.

Here, $\sigma^2_{u,v}$ is the variance of the $u,v$-th frequency dimension. We construct a mask, $M^{(c)}_{u,v}$, which only utilizes the top-$k$ dimensions based on the $\eta_{u,v}$ to maximally preserve the class-wise variance of $D$ as follows:

$$M^{(c)}_{u,v} = \begin{cases} 1 & \text{if } \eta_{u,v} \text{ is among the top-}k \text{ values} \\ 0 & \text{otherwise} \end{cases} \tag{4}$$

This masking strategy is efficient as it can be computed solely using the dataset statistics and does not require additional training with a deep neural network. This form of modeling is different from traditional filters such as low/high-pass filters, or band-stop filters, which are common choices in image processing. These frequency-range-based filters pass only certain frequency ranges by utilizing the stylized fact that each instance contains varying amounts of information across different frequency ranges. We conjecture that this type of class-agnostic filter cannot capture the discriminative features of each class, thereby failing to provide adequate information for downstream tasks, i.e. classification. This claim is supported by our empirical studies in Section 4.3.

**Inverse Frequency Transform $\mathcal{F}^{-1}$.**    We utilize the inverse frequency transform, $\mathcal{F}^{-1}$, to transform the inferred frequency representation to the corresponding instance on the spatial domain. The characteristics of the inverse frequency transform make it highly suitable as a choice for dataset distillation. First, $\mathcal{F}^{-1}$ is a differentiable function that enables the back-propagation of the gradient to the corresponding frequency domain. Second, $\mathcal{F}^{-1}$ is a static and invariant function, which does not require an additional parameter. Therefore, $\mathcal{F}^{-1}$ does not require any budget and training that could lead to inefficiency and instability. Third, the transformation process is efficient and fast. For $d$-dimension input vector, $\mathcal{F}^{-1}$ requires $\mathcal{O}(d \log d)$ operation while the convolution layer with $r$-size kernel needs $\mathcal{O}(dr)$. Therefore, in the common situation, where $\log d < r$, $\mathcal{F}^{-1}$ is faster than the convolution layer.[3] We provide an ablation study to compare the effectiveness of each frequency transform in Section 4.3 and Appendix D.8.

**Learning Framework.**    Following the tradition of dataset distillation [20, 26, 7], the training and evaluation stages of FreD are as follows:

$$F^* = \arg\min_F \mathcal{L}_{DD}(\tilde{S}, D) \text{ where } \tilde{S} = \mathcal{F}^{-1}(M \odot F) \qquad \text{(Training)} \tag{5}$$

$$\theta^* = \arg\min_\theta \mathcal{L}(\tilde{S}^*; \theta) \text{ where } \tilde{S}^* = \mathcal{F}^{-1}(M \odot F^*) \qquad \text{(Evaluation)} \tag{6}$$

---

[3]There are several researches which utilize this property to replace the convolution layer with a frequency transform [2, 12, 15].

---

**Algorithm 1** FreD: Frequency domain-based Dataset Distillation

---

1: **Input:** Original dataset $D$; Number of classes $C$; Frequency transform $\mathcal{F}$; Distillation loss $\mathcal{L}_{DD}$; Dimension budget per instance $k$; Number of frequency representations $|F|$; Learning rate $\alpha$
2: Initialize $F = \emptyset$
3: **for** $c = 1$ **to** $C$ **do**
4:     $F^{(c)} \leftarrow \{(\mathcal{F}(x^{(i)}), y^{(i)})\}_{i=1}^{\frac{|F|}{C}}$ from a class-wise mini-batch $\{(x^{(i)}, y^{(i)})\}_{i=1}^{\frac{|F|}{C}} \sim D^{(c)}$
5:     Initialize $M^{(c)}$ by Eq. (4) of the main paper
6: **end for**
7: **repeat**
8:     $F \leftarrow F - \alpha \nabla \mathcal{L}_{DD}(\mathcal{F}^{-1}(M \odot B_F), B_D)$ from a mini-batch $B_D \sim D$ and $B_F \sim F$
9: **until** convergence
10: **Output:** Masked frequency representations $M \odot F$

---

where $M \odot F$ denotes the collection of instance-wise masked representation i.e. $M \odot F = \{(M^{(y^{(i)})}(f^{(i)}), y^{(i)})\}_{i=1}^{|F|}$. By estimating $F^*$ from the training with Eq (5), we evaluate the effectiveness of $F^*$ by utilizing $\theta^*$, which is a model parameter inferred from training with the transformed dataset, $\tilde{S} = \mathcal{F}^{-1}(M \odot F^*)$. Algorithm 1 provides the instruction of FreD.

**Budget Allocation.** When we can store $n$ instances which is $d$-dimension vector for each class, the budget for dataset distillation is limited by $n \times d$. In FreD, we utilize $k < d$ dimensions for each instance. Therefore, we can accommodate $|F| = \lfloor n(\frac{d}{k}) \rfloor > n$ instances with the same budget. After the training of FreD, we acquire $M \odot F^*$, which actually shares the same dimension as the original image. However, we only count $k$ non-zero elements on each $f$ because storing 0 in $d - k$ dimension is negligible by small bytes. Please refer to Appendix E.3 for more discussion on budget allocation.

### 3.3 Theoretic Analysis of FreD

This section provides theoretical justification for the dimension selection of FreD by EVR, $\eta$. For validation, we assume that $\mathcal{F}$ is linearly bijective.

**Proposition 1.** *Let domain $A$ and $B$ be connected by a linear bijective function, $W$. The sum of $\eta$ over a subset of dimensions in domain $A$ for a dataset $X$ is equal to the sum of $\eta$ for the dataset transformed to domain $B$ using only the corresponding subset of dimensions.*

Proposition 1 claims that if two different domains are linearly bijective, the sum of EVR that utilizes only specific dimensions remains the same even when transformed into a different domain. In other words, if the sum of EVR of specific dimensions in the frequency domain is high, this value can be maintained when transforming a dataset only with those dimensions into the other domain.

**Corollary 1.** *Assume that two distinct domains, $B$ and $C$, are linearly bijective with domain $A$ by $W_B$ and $W_C$. let $X$ be a dataset in domain $A$, and $X_B$ and $X_C$ be the datasets transformed to domains $B$ and $C$, respectively. Let $V^*_{B,k}$ and $V^*_{C,k}$ be the set of $k$ dimension indexes that maximize $\eta$ in each domain. Let $\eta^*_{B,k}$ and $\eta^*_{C,k}$ be the corresponding sum of $\eta$ for each domain. If $\eta^*_{B,k} \geq \eta^*_{C,k}$, then the sum of $\eta$ for $W_{V^*_{B,k}} X_B$ is greater than that of $W_{V^*_{C,k}} X_C$.*

Corollary 1 claims that for multiple domains that have a bijective relationship with a specific domain, if a certain domain can obtain the high value of sum of EVR with the same number of dimensions, then when inverse transformed, it can improve the explanation in original domain. Based on the information concentration of the frequency domain, FreD can be regarded as a method that can better represent the distribution of the original dataset, $D$, based on $S$.

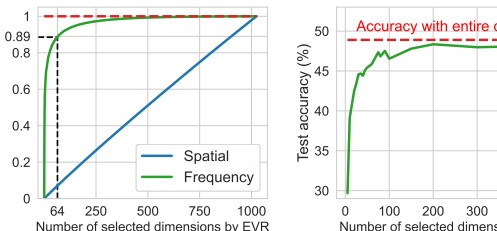

(a) Comparison of domain    (b) Efficiency of EVR selection

Figure 5: Empirical evidence for EVR-based selection.

PCA, which sets the principal components of the given dataset as new axes, can ideally preserve $\eta$.

Table 1: Test accuracies (%) on SVHN, CIFAR-10, and CIFAR-100. "IPC" denotes the number of images per class. "#Params" denotes the total number of budget parameters. The best results and the second-best result are highlighted in **bold** and underline, respectively.

| | | SVHN | | | CIFAR-10 | | | CIFAR-100 | | |
|---|---|---|---|---|---|---|---|---|---|---|
| | IPC | 1 | 10 | 50 | 1 | 10 | 50 | 1 | 10 | 50 |
| | #Params | 30.72k | 307.2k | 1536k | 30.72k | 307.2k | 1536k | 30.72k | 307.2k | 1536k |
| Coreset | Random | 14.6 ±1.6 | 35.1 ±4.1 | 70.9 ±0.9 | 14.4 ±2.0 | 26.0 ±1.2 | 43.4 ±1.0 | 4.2 ±0.3 | 14.6 ±0.5 | 30.0 ±0.4 |
| | Herding | 20.9 ±1.3 | 50.5 ±3.3 | 72.6 ±0.8 | 21.5 ±1.3 | 31.6 ±0.7 | 40.4 ±0.6 | 8.4 ±0.3 | 17.3 ±0.3 | 33.7 ±0.5 |
| Input-sized parameterization | DC | 31.2 ±1.4 | 76.1 ±0.6 | 82.3 ±0.3 | 28.3 ±0.5 | 44.9 ±0.5 | 53.9 ±0.5 | 12.8 ±0.3 | 25.2 ±0.3 | - |
| | DSA | 27.5 ±1.4 | 79.2 ±0.5 | 84.4 ±0.4 | 28.8 ±0.7 | 52.1 ±0.5 | 60.6 ±0.5 | 13.9 ±0.3 | 32.3 ±0.3 | 42.8 ±0.4 |
| | DM | - | - | - | 26.0 ±0.8 | 48.9 ±0.6 | 63.0 ±0.4 | 11.4 ±0.2 | 29.7 ±0.2 | 43.6 ±0.4 |
| | CAFE+DSA | 42.9 ±3.0 | 77.9 ±0.6 | 82.3 ±0.4 | 31.6 ±0.8 | 50.9 ±0.5 | 62.3 ±0.4 | 14.0 ±0.2 | 31.5 ±0.2 | 42.9 ±0.2 |
| | TM | 58.5 ±1.4 | 70.8 ±1.8 | 85.7 ±0.1 | 46.3 ±0.8 | 65.3 ±0.7 | 71.6 ±0.2 | 24.3 ±0.2 | 40.1 ±0.4 | 47.7 ±0.2 |
| | KIP | 57.3 ±0.1 | 75.0 ±0.1 | 80.5 ±0.1 | 49.9 ±0.2 | 62.7 ±0.3 | 68.6 ±0.2 | 15.7 ±0.2 | 28.3 ±0.1 | - |
| | FRePo | - | - | - | 46.8 ±0.7 | 65.5 ±0.4 | 71.7 ±0.2 | 28.7 ±0.1 | 42.5 ±0.2 | 44.3 ±0.2 |
| Parameterization | IDC | 68.1 ±0.1 | 87.3 ±0.2 | 90.2 ±0.1 | 50.0 ±0.4 | 67.5 ±0.5 | 74.5 ±0.1 | - | - | - |
| | HaBa | 69.8 ±1.3 | 83.2 ±0.4 | 88.3 ±0.1 | 48.3 ±0.8 | 69.9 ±0.4 | 74.0 ±0.2 | 33.4 ±0.4 | 40.2 ±0.2 | 47.0 ±0.2 |
| | FreD | **82.2** ±0.6 | **89.5** ±0.1 | **90.3** ±0.3 | **60.6** ±0.8 | **70.3** ±0.3 | **75.8** ±0.1 | **34.6** ±0.4 | **42.7** ±0.2 | **47.8** ±0.1 |
| Entire original dataset | | 95.4 ±0.1 | | | 84.8 ±0.1 | | | 56.2 ±0.3 | | |
| Increment of decoded instances | IDC | ×5 | ×5 | ×5 | ×5 | ×5 | ×5 | - | - | - |
| | HaBa | ×5 | ×5 | ×5 | ×5 | ×5 | ×5 | ×5 | ×5 | ×5 |
| | FreD | ×16 | ×8 | ×4 | ×16 | ×6.4 | ×4 | ×8 | ×2.56 | ×2.56 |

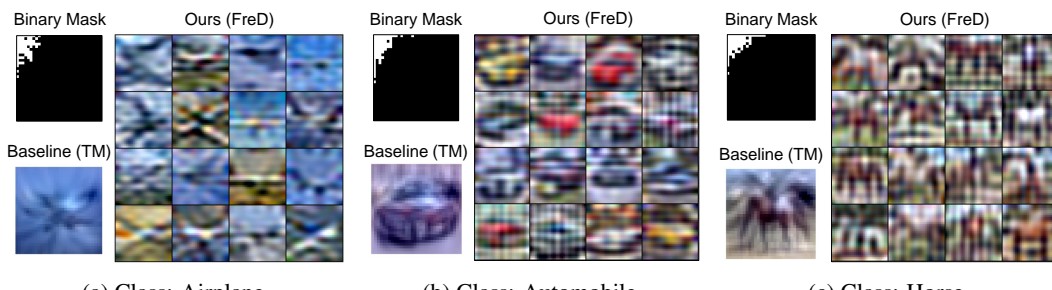

(a) Class: Airplane          (b) Class: Automobile          (c) Class: Horse

Figure 6: Visualization of the binary mask, condensed image from TM, and the transformed images of FreD on CIFAR-10 with IPC=1 (#Params=30.72k). Binary masks and trained synthetic images are all the same size. The size of mask and baseline image were enlarged for a better layout.

However, PCA, being a data-driven transform function, cannot be practically utilized as a method for dataset distillation. Please refer to Appendix E.1 for supporting evidence of this claim.

Figure 5a shows the sum of $\eta$ in descending order in both the spatial and frequency domains. First, we observe that as the number of selected dimensions in the frequency domain increases, frequency domain is much faster in converging to the total variance than the spatial domain. This result shows that the EVR-based dimension selection in the frequency domain is more effective than the selection in the spatial domain. In terms of training, Figure 5b also shows that the performance of a dataset constructed with a very small number of frequency dimensions converges rapidly to the performance of a dataset with the entire frequency dimension.

## 4 Experiments

### 4.1 Experiment Setting

We evaluate the efficacy of FreD on various benchmark datasets, i.e. SVHN [13], CIFAR-10, CIFAR-100 [8] and ImageNet-Subset [6, 3, 4]. Please refer to Appendix D for additional experimental results on other datasets. We compared FreD with both methods: 1) a method to model $S$ as an input-sized variable; and 2) a method, which parameterizes $S$, differently. As the learning with input-sized $S$, we chose baselines as DD [20], DSA [24], DM [25], CAFE+DSA [19], TM [3], KIP [14] and FRePo [27]. We also selected IDC [7] and HaBa [10] as baselines, which is categorized as the parameterization on $S$. we also compare FreD with core-set selection methods, such as random selection and herding [22]. We use trajectory matching objective (TM) [3] for $\mathcal{L}_{DD}$ as a default although FreD can use any dataset distillation loss. We evaluate each method by training 5 randomly initialized networks from scratch on optimized $S$. Please refer to Appendix C for a detailed explanation of datasets and experiment settings.

Table 2: Test accuracies (%) on CIFAR-10 under various dataset distillation loss and cross-architecture. "DC/DM/TM" denote the gradient/feature/trajectory matching for dataset distillation loss, respectively. We utilize AlexNet [9], VGG11 [18], and ResNet18 [5] for cross-architecture.

| | | DC | | | DM | | | TM | | |
|---|---|---|---|---|---|---|---|---|---|---|
| | IPC | 2 | 11 | 51 | 2 | 11 | 51 | 2 | 11 | 51 |
| | #Params | 61.44k | 337.92k | 1566.72k | 61.44k | 337.92k | 1566.72k | 61.44k | 337.92k | 1566.72k |
| ConvNet | Vanilla | 31.4 ±0.2 | 45.3 ±0.3 | 54.2 ±0.6 | 34.6 ±0.5 | 50.4 ±0.4 | 62.0 ±0.3 | 50.6 ±1.0 | 63.9 ±0.3 | 69.8 ±0.5 |
| | w/ IDC | 35.2 ±0.5 | 53.8 ±0.4 | 56.4 ±0.4 | 45.1 ±0.5 | 59.3 ±0.4 | 64.6 ±0.3 | 56.1 ±0.4 | 60.9 ±0.4 | 71.1 ±0.4 |
| | w/ HaBa | 34.1 ±0.5 | 49.9 ±0.5 | 58.9 ±0.2 | 37.3 ±0.1 | 56.8 ±0.1 | 64.4 ±0.4 | 56.8 ±0.4 | 69.5 ±0.3 | 73.3 ±0.2 |
| | w/ FreD | 45.3 ±0.5 | 55.8 ±0.4 | 59.8 ±0.5 | 55.9 ±0.4 | 61.3 ±0.8 | 66.6 ±0.6 | 61.4 ±0.3 | 70.7 ±0.5 | 75.5 ±0.2 |
| Average of Cross-Architectures | Vanilla | 22.0 ±0.9 | 29.2 ±0.9 | 34.1 ±0.6 | 21.5 ±2.2 | 39.5 ±1.1 | 52.6 ±0.7 | 33.1 ±1.1 | 43.9 ±1.4 | 55.0 ±1.0 |
| | w/ IDC | 28.7 ±1.2 | 35.4 ±0.6 | 40.2 ±0.7 | 37.3 ±1.1 | 50.5 ±0.6 | 61.3 ±0.5 | 42.5 ±1.5 | 48.7 ±1.8 | 61.5 ±1.0 |
| | w/ HaBa | 25.4 ±0.9 | 31.4 ±0.7 | 35.5 ±0.9 | 30.1 ±0.6 | 47.0 ±0.5 | 60.1 ±0.6 | 46.4 ±1.0 | 55.8 ±1.8 | 64.0 ±0.9 |
| | w/ FreD | 37.3 ±0.9 | 37.4 ±0.7 | 42.7 ±0.8 | 48.1 ±0.7 | 57.3 ±0.8 | 65.0 ±0.7 | 49.7 ±1.0 | 60.1 ±0.7 | 69.1 ±0.7 |

Table 3: Test accuracies (%) on ImageNet-Subset ($128 \times 128$) under IPC=2 (#Params=983.04k).

| Model | ImgNette | ImgWoof | ImgFruit | ImgYellow | ImgMeow | ImgSquawk |
|---|---|---|---|---|---|---|
| TM | 55.2 ±1.1 | 30.9 ±1.3 | 31.8 ±1.6 | 49.7 ±1.4 | 35.3 ±2.2 | 43.9 ±0.6 |
| w/ IDC | 65.4 ±1.2 | 37.6 ±1.6 | 43.0 ±1.5 | 62.4 ±1.7 | 43.1 ±1.2 | 55.5 ±1.2 |
| w/ HaBa | 51.9 ±1.7 | 32.4 ±0.7 | 34.7 ±1.1 | 50.4 ±1.6 | 36.9 ±0.9 | 41.9 ±1.4 |
| w/ FreD | 69.0 ±0.9 | 40.0 ±1.4 | 46.3 ±1.2 | 66.3 ±1.1 | 45.2 ±1.7 | 62.0 ±1.3 |

Table 4: Test accuracies (%) on 3D MNIST.

| IPC | 1 | 10 | 50 |
|---|---|---|---|
| #Params | 40.96k | 409.6k | 2048k |
| Random | 17.2 ±0.5 | 49.6 ±0.7 | 60.3 ±0.7 |
| DM | 42.5 ±0.9 | 58.6 ±0.8 | 64.7 ±0.5 |
| w/ IDC | 51.9 ±1.5 | 54.0 ±0.5 | 56.8 ±0.3 |
| w/ FreD | 54.9 ±0.5 | 62.9 ±0.5 | 66.6 ±0.7 |
| Entire dataset | | 78.7 ±1.1 | |

## 4.2 Experimental Results

**Performance Comparison.** Table 1 presents the test accuracies of the neural network, which is trained on $S$ inferred from each method. FreD achieves the best performances in all experimental settings. Especially, when the limited budget is extreme, i.e. IPC=1 (#Params=30.72k); FreD shows significant improvements compared to the second-best performer: 12.4%p in SVHN and 10.6%p in CIFAR-10. This result demonstrates that using the frequency domain, where information is concentrated in specific dimensions, has a positive effect on efficiency and performance improvement, especially in situations where the budget is very small. Please refer to Appendix D for additional experimental results on other datasets.

**Qualitative Analysis.** Figure 6 visualizes the synthetic dataset by FreD on CIFAR-10 with IPC=1 (#Params=30.72k). In this setting, we utilize 64 frequency dimensions per channel, which enables the construction of 16 images per class under the same budget. The results show that each class contains diverse data instances. Furthermore, despite of huge reduction in dimensions i.e. $64/1024 = 6.25\%$, each image contains class-discriminative features. We also provide the corresponding binary masks, which are constructed by EVR value. As a result, the low-frequency dimensions in the frequency domain were predominantly selected. It supports that the majority of frequency components for image construction are concentrated in the low-frequency region [1, 21]. Furthermore, it should be noted that our EVR-based mask construction does not enforce keeping the low-frequency components, what EVR only enforces is keeping the components with a higher explanation ratio on the image feature. Therefore, the constructed binary masks are slightly different for each class. Please refer to Appendix D.10 for more visualization.

**Compatibility of Parameterization.** The parameterization method in dataset distillation should show consistent performance improvement across different distillation losses and test network architectures. Therefore, we conduct experiments by varying the dataset distillation loss and test network architectures. In the case of HaBa, conducting an experiment at IPC=1 is structurally impossible due to the existence of a hallucination network. Therefore, for a fair comparison, we basically follow HaBa's IPC setting such as IPC=2,11,51. In Table 2, FreD shows the highest performance improvement for all experimental combinations. Specifically, FreD achieves a substantial performance gap to the second-best performer up to 10.8%p in training architecture and 10.6%p in cross-architecture generalization. Furthermore, in the high-dimensional dataset cases, Table 3 verifies that FreD consistently outperforms other parameterization methods. These results demonstrate that the frequency domain exhibits high compatibility and consistent performance improvement, regardless of its association with dataset distillation objective and test network architecture.

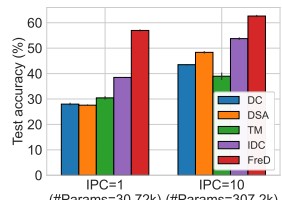

Figure 7: Test accuracies (%) on CIFAR-10-C.

Table 5: Test accuracies (%) on ImageNet-Subset-C. Note that the TM on ImgSquawk is excluded because the off-the-shelf synthetic dataset is not the default size of $128 \times 128$.

| #Params | Model | ImgNette-C | ImgWoof-C | ImgFruit-C | ImgYellow-C | ImgMeow-C | ImgSquawk-C |
|---------|-------|-----------|-----------|------------|-------------|-----------|-------------|
| 491520 (IPC=1) | TM | 38.0 ±1.6 | 23.8 ±1.0 | 22.7 ±1.1 | 35.6 ±1.7 | 23.3 ±1.1 | - |
| | w/ IDC | 34.5 ±0.6 | 18.7 ±0.4 | 28.5 ±0.9 | 36.8 ±1.4 | 22.2 ±1.2 | 26.8 ±0.5 |
| | w/ FreD | **51.2** ±0.6 | **31.0** ±0.9 | **32.3** ±1.4 | **48.2** ±1.0 | **30.3** ±0.3 | **45.9** ±0.6 |
| 4915200 (IPC=10) | TM | 50.9 ±0.7 | 30.9 ±0.7 | 32.3 ±0.8 | 45.6 ±1.0 | 30.1 ±0.5 | 44.4 ±1.8 |
| | w/ IDC | 40.4 ±1.0 | 21.9 ±0.3 | 32.2 ±0.7 | 39.6 ±0.5 | 23.9 ±0.8 | 40.5 ±0.7 |
| | w/ FreD | **55.2** ±0.8 | **33.8** ±0.8 | **35.7** ±0.6 | **47.9** ±0.4 | **31.3** ±0.9 | **52.5** ±0.8 |

**3D Point Cloud Dataset.** As the spatial dimension of the data increases, the required dimension budget for each instance also grows exponentially. To validate the efficay of FreD on data with dimensions higher than 2D, we assess FreD on 3D point cloud data, 3D MNIST.[4] Table 4 shows the test accuracies on the 3D MNIST dataset. FreD consistently achieves significant performance improvement over the baseline methods. This confirms the effectiveness of FreD in 2D image domain as well as 3D point cloud domain.

**Robustness against Corruption.** Toward exploring the application ability of dataset distillation, we shed light on the robustness against the corruption of a trained synthetic dataset. We utilize the following test datasets: CIFAR-10.1 and CIFAR-10-C for CIFAR-10, ImageNet-Subset-C for ImagNet-Subset. For CIFAR-10.1 and CIFAR-10-C experiments, we utilize the off-the-shelf synthetic datasets which are released by the authors of each paper. We report the average test accuracies across 15 types of corruption and 5 severity levels for CIFAR-10-C and ImageNet-Subset-C.

Figure 7 and Table 5 show the results of robustness on CIFAR-10-C and ImageNet-Subset-C, respectively. From both results, FreD shows the best performance over the whole setting which demonstrates the superior robustness against corruption. We want to note that IDC performs worse than the baseline in many ImageNet-Subset-C experiments (see Table 5) although it shows performance improvement on the ImageNet-Subset (see Table 3). On the other hand, FreD consistently shows significant performance improvement regardless of whether the test dataset is corrupted. It suggests that the frequency domain-based parameterization method shows higher domain generalization ability than the spatial domain-based parameterization method. Please refer to Appendix D.5 for the results of CIFAR-10.1 and detailed results based on corruption types of CIFAR-10-C.

To explain the rationale, corruptions that diminish the predictive ability of a machine learning model often occur at the high-frequency components. Adversarial attacks and texture-based corruptions are representative examples [11, 23]. Unlike FreD, which can selectively store information about an image's frequency distribution, transforms such as factorization or upsampling are well-known for not preserving frequency-based information well. Consequently, previous methods are likely to suffer a decline in predictive ability on datasets that retain class information while adding adversarial noise. In contrast, FreD demonstrates relatively good robustness against distribution shifts by successfully storing the core frequency components that significantly influence class recognition, regardless of the perturbations applied to individual data instances.

**Collaboration with Other Parameterization.** The existing method either performs resolution resizing in the spatial domain or uses a neural network to change the dimension requirement of the spatial domain. On the other hand, FreD optimizes the coefficient of the frequency domain dimension and transforms it into the spatial domain through the inverse frequency transform. Therefore, FreD can be applied orthogonally to the existing spatial domain-based parameterization methods. Table 6 shows the performance of different parameterizations applied to HaBa. From the results, we observed that FreD further enhances

Table 6: Test accuracies (%) of each collaboration on CIFAR-10.

| IPC #Params | 2 61.44k | 11 337.92k |
|-------------|----------|------------|
| TM | 50.6 ±1.0 | 63.9 ±0.3 |
| w/ HaBa | 56.8 ±0.4 | 69.5 ±0.3 |
| w/ IDC & HaBa | 61.3 ±0.3 | 70.9 ±0.4 |
| w/ FreD & HaBa | **62.3** ±0.1 | **72.9** ±0.2 |

the performance of HaBa. Furthermore, it is noteworthy that the performance of HaBa integrated with FreD is higher than the combination of IDC and HaBa. These results imply that FreD can be well-integrated with spatial domain-based parameterization methods.

---

[4] https://www.kaggle.com/datasets/daavoo/3d-mnist

## 4.3 Ablation Studies

**Effectiveness of Binary Mask $M$.** We conducted a comparison experiment to validate the explained variance ratio as a criterion for the selection of frequency dimensions. We selected the baselines for the ablation study as follows: Low-pass, Band-stop, High-pass, Random, and the magnitude of the amplitude in the frequency domain. We fixed the total budget and made $k$ the same. Figure 8a illustrates the ablation study on different variations of criterion for constructing $M$. We skip the high-pass mask because of its low performance:

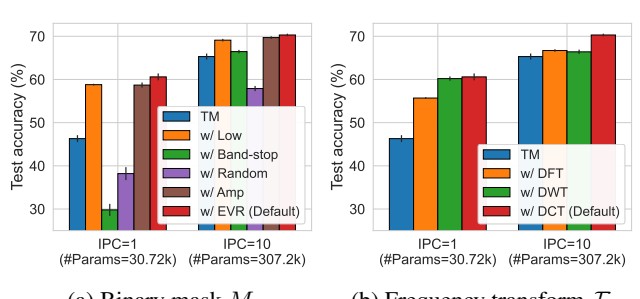

(a) Binary mask $M$        (b) Frequency transform $\mathcal{F}$

Figure 8: Ablation studies on (a) the binary mask $M$, and (b) the frequency transform $\mathcal{F}$.

14.32% in IPC=1 (#Params=30.72k) and 17.11% in IPC=10 (#Params=307.2k). While Low-pass and Amplitude-based dimension selection also improves the performance of the baseline, EVR-based dimension selection consistently achieves the best performance.

**Effectiveness of Frequency Transform $\mathcal{F}$.** We also conducted an ablation study on the frequency transform. Note that the FreD does not impose any constraints on the utilization of frequency transform. Therefore, we compared the performance of FreD when applying widely used frequency transforms such as the Discrete Cosine Transform (DCT), Discrete Fourier Transform (DFT), and Discrete Wavelet Transform (DWT). For DWT, we utilize the Haar wavelet function and low-pass filter instead of an EVR mask. As shown in Figure 8b, we observe a significant performance improvement regardless of the frequency transform. Especially, DCT shows the highest performance improvement than other frequency transforms. Please refer to Appendix D.8 for additional experiments and detailed analysis of the ablation study on frequency transform.

**Budget Allocation.** Dataset distillation aims to include as much information from the original dataset as possible on a limited budget. FreD can increase the number of data $|F|$ by controlling the dimension budget per instance $k$, and FreD stores the frequency coefficients selected by EVR as $k$-dimensional vector. For example, with a small value of $k$, more data can be stored i.e. large $|F|$. This is a way to increase the quantity of instances while decreasing the quality of variance and reconstruction. By utilizing this flexible trade-off, we can pick the balanced point between quantity and quality to further increase the efficiency of our limited budget. Figure 9 shows the performance of selecting the dimension budget per channel under different budget situations. Note that, for smaller budgets i.e. IPC=1 (#Params=30.72k), increasing $|F|$ performs better. For larger budget cases, such as IPC=50 (#Params=1536k), allocating more dimensions to each instance performs better i.e. large $k$. This result shows that there is a trade-

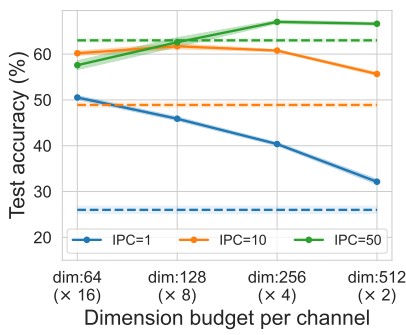

Figure 9: Ablation study on budget allocation of FreD with DM. Dashed line indicates the performance of DM.

off between the quantity and the quality of data instances depending on the budget size.

## 5 Conclusion

This paper proposes a new parameterization methodology, FreD, that utilizes the augmented frequency domain. FreD selectively utilizes a set of dimensions with a high variance ratio in the frequency domain, and FreD only optimizes the frequency representations of the corresponding dimensions in the junction with the frequency transform. Based on the various experiments conducted on benchmark datasets, the results demonstrate the efficacy of utilizing the frequency domain in dataset distillation. Please refer to Appendix G for the limitation of the frequency domain-based dataset distillation.

## Acknowledgement

This work was supported by the National Research Foundation of Korea (NRF) grant funded by the Korea government (MSIT) (No.2021R1A2C200981613). ※ MSIT: Ministry of Science and ICT

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
