# Appendix A    Literature Reviews on Related Works

## A.1    Dataset Distillation

In this section, we briefly review the methodology of constructing $S$ as an input-sized vector and provide a detailed review of our main comparative methods, HaBa [19], IDC [14] and GLaD [4].

**Input-sized Parameterization.**    Dataset Distillation (DD) [33] aims at finding the synthetic dataset $S$ with a bi-level optimization. The main idea of bi-level optimization is that the network parameter $\theta_S$, which is trained on $S$, minimizes the (population) risk of the original dataset $D$. Dataset Condensation (DC) [45] introduces a proxy objective, which aims at matching the layer-wise gradients of a network over the optimization path of $S$. Differentiable Siamese Augmentation (DSA) [43] applies the differentiable and identical data augmentation to original data instances and synthetic data instances at each training step. Contrary to gradient matching i.e. short-range trajectory matching [45], Trajectory Matching (TM) aims at transferring the knowledge of long-range trajectory from pre-trained with the original dataset. It minimizes the difference between the training trajectory on synthetic data and the training trajectory on real data. Distribution Matching (DM) [44] points out the computation cost of precedents. Therefore, the authors propose a new objective that aims at aligning the feature distributions of both the original dataset and the synthetic dataset within sampled embedding spaces. CAFE [32] extends the DM by layer-wise feature matching. Kernel Inducing Point (KIP) [24] introduces a kernel-based objective that leverages infinite-width neural networks. It optimizes to let condensed datasets be kernel inducing points in kernel ridge regression. FRePo [46] points out the meta-gradient computation and overfitting in dataset distillation. FRePo overcomes these challenges by utilizing the truncated backpropagation through time and model pool.

**HaBa [19].**    HaBa proposed a technique for dataset factorization, which involves breaking the synthetic dataset into bases and hallucinator networks. The hallucinator takes bases as input and generates image instances. By learning bases and hallucinators, the resulting model could produce more diverse samples based on the available budget. However, incorporating an additional network in distillation requires a separate budget, which is distinct from the data instances. For example, HaBa does not perform when 1 image per class setting, although using light-weight hallucinators. Furthermore, joint learning of both the network and data usually leads to instability in the training.

**IDC [14].**    IDC configures the synthetic dataset as several reduced-size of spatial images and utilizes the differentiable multi-formation function to restore to the original size. The usual choice of multi-formation function is an upsampling module, which does not require training. Therefore, efficient parameterization enables the increasing the available number of data instances. However, the compression process still takes place in the spatial domain, leading to the loss of information and inefficient utilization of the budget. Additionally, [8] empirically showed that upsampling methods cause distortion in the spectral distribution of natural images.

**GLaD [4].**    GLaD employs a pre-trained generative model and distills the synthetic dataset in the latent space of the generative model, such as Generative Adversarial Networks (GAN). By leveraging the generative model, GLaD encourages better generalization to unseen architecture and scale to high-dimensional datasets. However, generative models typically require a large number of parameters, which introduces several inefficiencies. Firstly, storing a generative model with many parameters is burdensome in dataset distillation which is budget-constrained. Due to the budget constraint, GLaD proposes to use a generative model for training, and at the end of training, create a distilled instance by combining the distilled latent code and generative model to create a distilled instance and store it in the budget. This eliminates the need to allocate a budget for the generative model, but the amount of budget occupied by one distilled instance is the same as the input-sized parameterization method. As a result, it creates the same number of instances as the input-sized parameterization method, which makes the synthetic dataset insufficiently expressive. There is also a computational inefficiency because it takes more time to move forward and backward due to the large number of parameters. Finally, Frequency transform is dataset agnostic, while the deep generative model needs to apply a suitable structure to the dataset. This has the inefficiency of selecting the appropriate structure based on the dataset.

## A.2 Frequency Transform

**Additional Review of Frequency Transform.**   As mentioned in the main paper, the form of the frequency transform depends on the selection of the basis function $\phi(a, b, u, v)$ (see Eq. (2) in the main paper). Discrete Cosine Transform (DCT) uses the cosine function as the basis function i.e. $\phi(a, b, u, v) = cos\left(\frac{\pi}{d_1}(a + \frac{1}{2}u)\right)cos\left(\frac{\pi}{d_2}(b + \frac{1}{2}v)\right)$. Discrete Fourier Transform (DFT) utilizes the exponential function as the basis function i.e. $\phi(a, b, u, v) = e^{-i2\pi(\frac{ua}{d_1} + \frac{vb}{d_2})}$. Discrete Wavelet Transform (DWT) employs the wavelet function, such as the Haar wavelet function or the Daubechies wavelet. In the case of images with multiple color channels, both frequency transform and inverse frequency transform can be independently applied to each channel. There are various research areas in machine learning, which use the property of frequency domain. In the following paragraphs, we review research conducted in the direction of utilizing the property of frequency domain such as adversarial attacks and analyze the neural network.

**Adversarial Attack.**   Recently, in adversarial attack areas, there has been a discussion suggesting that attacks in the frequency domain exhibit higher effectiveness compared to attacks based in the spatial domain [9, 29]. In [9], the authors propose a method to constrain the search space of adversarial attacks to the low-frequency domain. This method consistently reduces the black-box attack's query cost. Furthermore, the authors of [29] show empirical evidence of the effectiveness of the low-frequency attack.

**Analyzing Neural Network.**   There are a bunch of studies that analyze neural networks in terms of frequency transforms. Spectral bias in deep neural networks [25, 2] is a well-known problem in machine learning, which describes the tendency of the network to prefer specific frequency components over other components while training. The presence of spectral bias in a deep neural network can have a significant impact on its ability to generalize to new data instances by restricting its ability to capture crucial patterns or features for a given task [2, 37]. To prevent such biased training, [42, 39] designed a network and the corresponding loss function that takes transformed values in the frequency domain as input. To prevent spectral bias during the training, [36, 13] introduced frequency-based regularization techniques, while [20] proposed augmentation methods based on the frequency domain.

# Appendix B   Proofs of Theoretical Evidences

## B.1   Proof of Proposition 1

**Proposition 1.** *Let domain $A$ and $B$ be connected by a linear bijective function, $W$. The sum of $\eta$ over a subset of dimensions in domain $A$ for a dataset $X$ is equal to the sum of $\eta$ for the dataset transformed to domain $B$ using only the corresponding subset of dimensions.*

*Proof.* Mathematically, let $X$ be the $d$-dimensional dataset with $n$ samples in the domain $A$, and let $X_B$ be the transformed dataset in the domain $B$. Let $S \subseteq \{1, 2, ..., d\}$ be the subset of dimensions for which we want to calculate the sum of explained variance ratio. Then, the sum of explained variance ratio for $S$ in the domain $A$ is given by:

$$R_A(S) = \frac{\sum_{i \in S} \lambda_i}{\sum_{i=1}^{d} \lambda_i} \tag{1}$$

where $\lambda_i$ is the eigenvalues of $i$-th dimension of the covariance matrix of $X$. As noted in the assumption, bijective function $W$ exists to transform the $X$ into $X_B$, i.e. $X_B = WX$. We can write the covariance matrix of $X_B$ as:

$$\Sigma_B = \frac{1}{n}X_B X_B^T = \frac{1}{n}(WX)(WX)^T = W(\frac{1}{n}XX^T)W^T = W\Sigma_A W^T \tag{2}$$

where $\Sigma_A$ is the covariance matrix of $X$ in the domain $A$. Having said that, let $S_B = \{j \,|\, j = W(i), i \in S\}$ be the corresponding subset of dimensions in the domain $B$. It should be noted that each element in $S_B$ do not have to be one-hot dimension. Also, in a linear bijective transformation,

orthogonality in the original space is preserved in the transformed space. Then, the sum of explained variance ratios for the dimension subset, $S_B$, in the domain $B$ is given as follows:

$$R_B(S_B) = \frac{\sum_{j' \in S_B} \lambda_j'}{\sum_{j=1}^d \lambda_j} \tag{3}$$

where $\lambda_j$ is an eigenvalue of $j$-th dimension of the covariance matrix of $X_B$. Now, we can show that $R_A(S) = R_B(S_B)$ as follows:

$$R_B(S_B) = \frac{\sum_{j \in S_B} \lambda_j}{\sum_{j=1}^d \lambda_j} = \frac{\sum_{j \in S_B} \lambda_{W(i)}}{\sum_{j=1}^d \lambda_{W(i)}} = \frac{\sum_{i \in S} \lambda_i}{\sum_{i=1}^d \lambda_i} = R_A(S) \tag{4}$$

where we used the fact that the eigenvalues of the covariance matrix are the same for $X$ and $X_B$ (i.e., $\lambda_i = \lambda_{W(j)}$ for all $i, j$), and the fact that the sum of eigenvalues is invariant under bijective linear transformation. □

Therefore, we have shown that the sum of the explained variance ratio for a subset of dimensions in the domain $A$ is the same as the explained variance ratio sum for the domain $B$ when transforming the domain $A$ dataset to the domain $B$ using only that subset of dimensions. We re-arrange the claim as follows: The sum of explained variance ratios of a masked dataset for a specific dimension subset remains preserved even under linearly bijective transformations between domains.

## B.2 Proof of Corollary 1

**Corollary 1.** *Assume that two distinct domains, $B$ and $C$, are linearly bijective with domain $A$ by $W_B$ and $W_C$. let $X$ be a dataset in domain $A$, and $X_B$ and $X_C$ be the datasets transformed to domains $B$ and $C$, respectively. Let $V_{B,k}^*$ and $V_{C,k}^*$ be the set of $k$ dimension indexes that maximize $\eta$ in each domain. Let $\eta_{B,k}^*$ and $\eta_{C,k}^*$ be the corresponding sum of $\eta$ for each domain. If $\eta_{B,k}^* \geq \eta_{C,k}^*$, then the sum of $\eta$ for $W_{V_{B,k}^*} X_B$ is greater than that of $W_{V_{C,k}^*} X_C$.*

*Proof.* In Proposition 1, we proved that the sum of explained variance ratios of a masked dataset for a specific dimension subset remains preserved even under linearly bijective transformations between domains. Having said that, $W_{V_{C,k}^*} X_C$ is a transformed dataset of $X_C$ from domain $C$ to domain $A$, where only the top-$k$ dimensions that maximize the sum of explained variance ratios are utilized. Proposition 1 states that this transformation preserves the sum of explained variance ratios, and since $\eta_{B,k}^* \geq \eta_{C,k}^*$, the sum of explained variance ratios is preserved even in terms of the relative magnitude between the explained variance ratios sum of the transformed datasets. □

# Appendix C  Experimental Details

## C.1  Dataset

In this paper, we evaluate FreD on a variety of benchmark datasets, including those widely used in dataset distillation.

- MNIST [18] is a handwritten digit image dataset with 60,000 images for training and 10,000 images for testing. Each image is a $28 \times 28$ gray-scale image and is categorized into 10 classes (digits from 0 to 9).
- Fashion MNIST [35] contains various fashion items images such as clothing and shoe. It consists of a training set of 60,000 grayscale images and a test set of 10,000 images. Each image has a $28 \times 28$ size. Fashion MNIST has 10 classes in total.
- SVHN [23] is a real-world digit image dataset with 73,257 images for training and 26,032 images for testing. Each image in the dataset is a $32 \times 32$ RGB image and belongs to one of 10 classes ranging from 0 to 9.
- CIFAR-10 [15] consists of $32 \times 32$ RGB images with 50,000 images for training and 10,000 images for testing. It has 10 classes in total and each class contains 5,000 images for training and 1,000 images for testing.

- CIFAR-100 [15] comprises a total of 60,000 $32 \times 32$ RGB images distributed across 100 classes. Within each class, 500 images are allocated for training, while 100 images are for testing. These 100 classes are further grouped into 20 superclasses, with each superclass consisting of 5 classes at a more specific level.
- 3D MNIST (https://www.kaggle.com/datasets/daavoo/3d-mnist) consists of 10,000 training data and 1,000 test data, and each data has $1 \times 16 \times 16 \times 16$ size. Each data instance is categorized into 10 classes.
- Tiny-ImageNet [17] is a downsampled subset of ImageNet [5] to a size of $64 \times 64$. This dataset consists of 200 classes and each class contains 500 images for training and 100 images for testing.
- ImageNet-Subset is a dataset consisting of a subset of similar features in the ImageNet [5]. By following the previous work, we consider diverse types of subsets: ImageNette (various objects)[12], ImageWoof (dog breeds)[12], ImageFruit (fruits) [3], ImageMeow (cats) [3], ImageSquawk (birds) [3], ImageYellow (yellowish things) [3], and ImageNet-[A, B, C, D, E] (based on ResNet50 performance) [4]. Each subset has 10 classes. We consider two types of resolution: $128 \times 128$ and $256 \times 256$.
- LSUN [41] aims at understanding the large-scale scene images. The original LSUN dataset has 10 classes and each class contains a large number of images, ranging from 120k to 3,000k for training. We consider two datasets, coined as LSUN-10k/LSUN-25k, which randomly sampled 10k/25k instances per class which resulted in a total 100k/250k instances, respectively. We also downsize each instance to a $128 \times 128$ size.
- CIFAR-10.1 [27] consists of 2,000 new test images which have same classes as CIFAR-10.
- CIFAR-10-C and ImageNet-C [11] aim at measuring the robustness of object recognition based on CIFAR-10 and ImageNet, respectively. They have 15 types of corruption and each corruption has five levels with level 5 indicating the most severest. We create ImageNet-Subset-C by selecting data from ImageNet-C that matches the ImageNet-Subset classes.

## C.2 Architecture

For 2D image datasets, we basically employ an $n$-depth convolutional neural network, coined ConvNetD$n$, by following the previous works. The ConvNetD$n$ has $n$ duplicate blocks, which consist of a convolution layer with $3 \times 3$-shape 128 filters, an instance normalization layer [31], ReLU, and an average pooling with $2 \times 2$ kernel size with stride 2. After the convolution blocks, a linear classifier outputs the logits. We utilize a different number of blocks depending on the resolution: ConvNetD3 for $28 \times 28$ and $32 \times 32$, ConvNetD4 for $64 \times 64$, ConvNetD5 for $128 \times 128$ and ConvNetD6 for $256 \times 256$. For the performance comparison for different test network architectures, we also follow the precedent: ResNet [10], VGG [30], AlexNet [16], and ViT [7].

For the 3D point cloud dataset; 3D MNIST, we implement a 3D version of ConvNet, coined Conv3DNet. Similarly, Conv3DNet has three duplicate blocks; a convolution layer with $3 \times 3 \times 3$-shape 64 filters, a 3D instance normalization, ReLU, and a 3D average pooling with $2 \times 2 \times 2$ with stride 2. A linear layer follows these convolution blocks.

## C.3 Implementation Configurations

We use trajectory matching objective (TM) [3] for $\mathcal{L}_{DD}$ as a default although FreD can use any dataset distillation loss. Similarly, we utilize Discrete Cosine Transform (DCT) as a default frequency transform $\mathcal{F}$. For the implementation of frequency transform, we utilize the open-source PyTorch library; **torch-dct** (https://github.com/zh217/torch-dct) for DCT and **pytorch_wavelets** (https://github.com/fbcotter/pytorch_wavelets) for Discrete Wavelet Transform (DWT). We utilize the built-in function of PyTorch for the Discrete Fourier Transform (DFT). We separately apply the frequency transform to each channel for RGB image datasets. We use an SGD optimizer with a momentum rate of 0.5 for all our experiments. Each experiment is trained with 15,000 iterations. Contrary to previous research [3, 19], FreD does not use the ZCA Whitening. We used four RTX 3090 GPUs by default and two Tesla A100 GPUs for CIFAR-100, Tiny-ImageNet, and ImageNet-Subset. We basically follow the evaluation protocol of the previous works [45, 44, 3]. We evaluate each method by training 5 randomly initialized networks from scratch on optimized $S$. We provide the detailed hyper-parameters in Table 9 (see the end of Appendix).

# Appendix D  Additional Experimental Results

## D.1  Performance Comparison on Low-dimensional Datasets

We evaluate our proposed method on low-dimensional datasets ($\leq 64 \times 64$ resolution) such as MNIST, Fashion MNIST, and Tiny-ImageNet. Table 1 shows that FreD achieves improved or competitive performances in most experimental settings. These results repeatedly support our conjecture: the utilization of the frequency domain yields beneficial outcomes in terms of enhancing performance.

FreD's motivation lies in leveraging select important dimensions of the frequency domain, which can contain much of the spatial domain's information, to utilize the given memory budget more efficiently. This efficiency manifests greater utility when the available memory budget is more limited. Through extensive experiments results, FreD demonstrates more substantial performance improvement in most experiments with an IPC=1 setting. TinyImageNet is originally a dataset with 500 instances per class, and the IPC=50 setting for this dataset could be considered a not-so-drastic reduction. In situations where such a significant reduction doesn't occur, FreD's motivation may be weakened. Excluding this particular setting, FreD consistently demonstrates performance improvement compared to the baseline across evaluations.

Table 1: Test accuracies (%) on MNIST, Fashion MNIST, and Tiny-ImageNet. The best results and the second-best result are highlighted in **bold** and underline, respectively. Note that IDC does not provide the standard deviation on MNIST and Fashion MNIST experiments in the original paper.

|  |  | MNIST | | Fashion MNIST | | Tiny-ImageNet | | |
| --- | --- | --- | --- | --- | --- | --- | --- | --- |
| | IPC | 1 | 10 | 1 | 10 | 1 | 10 | 50 |
| | #Params | 7.84k | 78.4k | 7.84k | 78.4k | 2457.6k | 24576k | 122880k |
| Coreset | Random | 64.9 ±3.5 | 95.1±0.9 | 51.4 ±3.8 | 73.8 ±0.7 | 1.4 ±0.1 | 5.0 ±0.2 | 15.0 ±0.4 |
| | Herding | 89.2 ±1.6 | 93.7 ±0.3 | 67.0 ±1.9 | 71.1 ±0.7 | 2.8 ±0.2 | 6.3 ±0.2 | 16.7 ±0.3 |
| Input-sized parameterization | DC | 91.7 ±0.5 | 97.4 ±0.2 | 70.5 ±0.6 | 82.3 ±0.4 | - | - | - |
| | DSA | 88.7 ±0.6 | 97.8 ±0.1 | 70.6 ±0.6 | 84.6 ±0.3 | - | - | - |
| | DM | 89.7 ±0.6 | 97.5 ±0.1 | - | - | 3.9 ±0.2 | 12.9 ±0.4 | 24.1 ±0.2 |
| | CAFE+DSA | 90.8 ±0.5 | 97.5 ±0.1 | 73.7 ±0.7 | 83.0 ±0.3 | - | - | - |
| | TM | 88.7 ±1.0 | 96.6 ±0.4 | 75.7 ±1.5 | 88.4 ±0.4 | 8.8 ±0.3 | 23.2 ±0.2 | **28.0** ±0.2 |
| | KIP | 90.1 ±0.1 | 87.5 ±0.0 | 73.5 ±0.5 | 86.8 ±0.1 | - | - | - |
| | FRePo | 93.0 ±0.4 | **98.6** ±0.1 | 75.6 ±0.3 | 86.2 ±0.2 | 15.4 ±0.3 | **25.4** ±0.2 | - |
| Parameterization | IDC | 94.2 | 98.4 | 81.0 | 86.0 | - | - | - |
| | HaBa | 92.4 ±0.4 | 97.4 ±0.2 | 80.9 ±0.7 | 88.6 ±0.2 | - | - | - |
| | FreD | **95.8** ±0.2 | 97.6 ±0.8 | **84.6** ±0.2 | **89.1** ±0.2 | **19.2** ±0.4 | 24.2 ±0.4 | 26.4 ±0.4 |
| Entire original dataset | | 99.6 ±0.0 | | 93.5 ±0.1 | | 37.6 ±0.4 | | |

## D.2  Performance Comparison on High-dimensional Datasets

We further evaluate our proposed method on high-dimensional datasets ($\geq 128 \times 128$ resolution). Table 2 and 3 present the results of extensive experiments on $128 \times 128$ resolution ImageNet-Subset. As in the case of low-dimensional datasets, FreD consistently achieves the highest performance improvement among the parameterization methods in most experimental settings. Since the performance of FreD at IPC=10 (#Params=4915.2k) already overwhelms the performance of HaBa of IPC=11 (#Params=5406.72k), we did not conduct the experiment of FreD on IPC=11 (#Params=5406.72k). Furthermore, in Table 4, FreD repeatedly shows better performance on $256 \times 256$ resolution ImageNet-Subset.

It should be noted that FreD significantly improves the performance of cross-architecture generalization. For instance, GLaD also improves cross-architecture performance, but it shows the performance degradation in the architecture used for training when the utilized dataset distillation loss is TM. On the other hand, FreD shows the best performance in all experiments. It means that FreD provides insight into how well the frequency domain-based parameterization method understands the task, rather than overfitting to a particular architecture.

In summary, these extensive experimental results continuously demonstrate the efficacy of utilizing the frequency domain in dataset distillation regardless of the image's resolution.

Table 2: Test accuracies (%) on ImageNet-Subset (Image-[Nette, Woof, Fruit, Yellow, Meow, Squawk], $128 \times 128$). Note that HaBa is structurally disabled to experiment in IPC=1 (#Params=491.52k) due to the nature of its methodology.

| #Params | Model | ImageNette | ImageWoof | ImageFruit | ImageYellow | ImageMeow | ImageSquawk |
|---|---|---|---|---|---|---|---|
| 491.52k (IPC=1) | TM | 47.7 ±0.9 | 28.6 ±0.8 | 26.6 ±0.8 | 45.2 ±0.8 | 30.7 ±1.6 | 39.4 ±1.5 |
| | w/ IDC | 61.4 ±1.0 | 34.5 ±1.1 | 38.0 ±1.1 | 56.5 ±1.8 | 39.5 ±1.5 | 50.2 ±1.5 |
| | w/ HaBa | - | - | - | - | - | - |
| | w/ FreD | **66.8** ±0.4 | **38.3** ±1.5 | **43.7** ±1.6 | **63.2** ±1.0 | **43.2** ±0.8 | **57.0** ±0.8 |
| 983.04k (IPC=2) | TM | 55.2 ±1.1 | 30.9 ±1.3 | 31.6 ±1.6 | 49.7 ±1.4 | 35.3 ±2.2 | 43.9 ±0.6 |
| | w/ IDC | 65.4 ±1.2 | 37.6 ±1.6 | 43.0 ±1.5 | 62.4 ±1.7 | 43.1 ±1.2 | 55.5 ±1.2 |
| | w/ HaBa | 51.9 ±1.7 | 32.4 ±0.7 | 34.7 ±1.1 | 50.4 ±1.6 | 36.9 ±0.9 | 41.9 ±1.4 |
| | w/ FreD | **69.0** ±0.9 | **40.0** ±1.4 | **46.3** ±1.2 | **66.3** ±1.1 | **45.2** ±1.7 | **62.0** ±1.3 |
| 4915.2k (IPC=10) | TM | 63.0 ±1.3 | 35.8 ±1.8 | 40.3 ±1.3 | 60.0 ±1.5 | 40.4 ±2.2 | 52.3 ±1.0 |
| | w/ IDC | 70.8 ±0.5 | 39.8 ±0.9 | 46.3 ±1.4 | 68.7 ±0.8 | 47.9 ±1.4 | 65.4 ±1.2 |
| | w/ HaBa | - | - | - | - | - | - |
| | w/ FreD | **72.0** ±0.8 | **41.3** ±1.2 | **47.0** ±1.1 | **69.2** ±0.6 | **48.6** ±0.4 | **67.3** ±0.8 |
| 5406.72k (IPC=11) | TM | 63.9 ±0.5 | 36.6 ±0.8 | 40.1 ±1.9 | 60.4 ±1.5 | 41.0 ±1.5 | 54.6 ±1.0 |
| | w/ HaBa | 64.7 ±1.6 | 38.6 ±1.3 | 42.5 ±1.6 | 63.0 ±1.6 | 42.9 ±0.9 | 56.8 ±1.0 |

Table 3: Test accuracies (%) on ImageNet-Subset (ImageNet-[A, B, C, D, E], $128 \times 128$) with IPC=1 (#Params=491.52k). "Cross" denotes the average test accuracy of trained AlexNet, VGG11, ResNet18, and ViT on each synthetic dataset.

| | ImageNet-A | | ImageNet-B | | ImageNet-C | | ImageNet-D | | ImageNet-E | |
|---|---|---|---|---|---|---|---|---|---|---|
| | ConvNet | Cross | ConvNet | Cross | ConvNet | Cross | ConvNet | Cross | ConvNet | Cross |
| DC | 43.2 ±0.6 | 38.7 ±4.2 | 47.2 ±0.7 | 38.7 ±1.0 | 41.3 ±0.7 | 33.3 ±1.9 | 34.3 ±1.5 | 26.4 ±1.1 | 34.9 ±1.5 | 27.4 ±0.9 |
| w/ GLaD | 44.1 ±2.4 | 41.8 ±1.7 | 49.2 ±1.1 | 42.1 ±1.2 | 42.0 ±0.6 | 35.8 ±1.4 | 35.6 ±0.9 | 28.0 ±0.8 | 35.8 ±0.9 | 29.3 ±1.3 |
| w/ FreD | **53.1** ±1.0 | **48.0** ±1.4 | **54.8** ±1.2 | **47.6** ±1.5 | **54.2** ±1.2 | **47.8** ±1.2 | **42.8** ±1.0 | **36.3** ±1.4 | **41.0** ±1.1 | **35.0** ±1.1 |
| DM | 39.4 ±1.8 | 27.2 ±1.2 | 40.9 ±1.7 | 24.4 ±1.1 | 39.0 ±1.3 | 23.0 ±1.4 | 30.8 ±0.9 | 18.4 ±0.7 | 27.0 ±0.8 | 17.7 ±0.9 |
| w/ GLaD | 41.0 ±1.5 | 31.6 ±1.4 | 42.9 ±1.9 | 31.3 ±3.9 | 39.4 ±0.7 | 26.9 ±1.2 | 33.2 ±1.4 | 21.5 ±1.0 | 30.3 ±1.3 | 20.4 ±0.8 |
| w/ FreD | **58.0** ±1.7 | **48.7** ±1.5 | **58.6** ±1.3 | **47.5** ±1.5 | **55.6** ±1.4 | **47.1** ±1.0 | **46.3** ±1.2 | **35.9** ±2.0 | **45.0** ±1.8 | **32.1** ±1.6 |
| TM | 51.7 ±0.2 | 33.4 ±1.5 | 53.3 ±1.0 | 34.0 ±3.4 | 48.0 ±0.7 | 31.4 ±3.4 | 43.0 ±0.6 | 27.7 ±2.7 | 39.5 ±0.9 | 24.9 ±1.8 |
| w/ GLaD | 50.7 ±0.4 | 39.9 ±1.2 | 51.9 ±1.3 | 39.4 ±1.3 | 44.9 ±0.4 | 34.9 ±1.1 | 39.9 ±1.7 | 30.4 ±1.5 | 37.6 ±0.7 | 29.0 ±1.1 |
| w/ FreD | **67.7** ±1.0 | **51.9** ±1.1 | **69.3** ±1.2 | **50.7** ±1.2 | **63.6** ±2.0 | **48.4** ±1.1 | **54.4** ±1.0 | **39.2** ±1.4 | **55.4** ±1.7 | **39.8** ±1.1 |

Table 4: Test accuracies (%) on ImageNet-Subset (ImageNet-[A, B, C, D, E], $256 \times 256$) with IPC=1 (#Params=1966.08k). "Cross" denotes the average test accuracy of trained AlexNet, VGG11, ResNet18, and ViT on each synthetic dataset.

| | ImageNet-A | | ImageNet-B | | ImageNet-C | | ImageNet-D | | ImageNet-E | |
|---|---|---|---|---|---|---|---|---|---|---|
| | ConvNet | Cross | ConvNet | Cross | ConvNet | Cross | ConvNet | Cross | ConvNet | Cross |
| DC | - | 38.3 ±4.7 | - | 32.8 ±4.1 | - | 27.6 ±3.3 | - | 25.5 ±1.2 | - | 23.5 ±2.4 |
| w/ GLaD | - | 37.4 ±5.5 | - | 41.5 ±1.2 | - | 35.7 ±4.0 | - | 27.9 ±1.0 | - | 29.3 ±1.2 |
| w/ FreD | 54.8 ±0.9 | **48.0** ±0.9 | 56.2 ±1.0 | **48.2** ±1.7 | 53.5 ±1.4 | **47.3** ±1.0 | 41.6 ±1.2 | **37.8** ±1.0 | 39.1 ±1.5 | **33.4** ±1.2 |

### D.3 Performance Comparison on Large-size Dataset

Distilling the dataset into a small cardinality synthetic dataset can be more effective when the size of the original is large. Therefore, we further investigate the usefulness of our method and several baselines on a dataset with a large number of instances. We choose LSUN dataset [41] as the large-size dataset. Table 5 provides performances of FreD and other baselines on the LSUN dataset. As a result, FreD achieves the best performance compared to the implemented baselines.

Table 5: Test accuracies (%) on LSUN.

| | LSUN-10k | | LSUN-25k | |
|---|---|---|---|---|
| | DC | DM | DC | DM |
| Vanilla | 24.0 ±1.1 | 22.3 ±0.4 | 23.9 ±0.5 | 22.3 ±0.4 |
| w/ IDC | 22.7 ±0.3 | 27.4 ±0.8 | 22.7 ±0.7 | 27.1 ±0.4 |
| w/ FreD | **30.3** ±0.9 | **37.1** ±0.2 | **32.1** ±0.2 | **36.3** ±0.6 |
| Entire dataset | 71.8 ±0.3 | | 72.8 ±0.3 | |

## D.4    More Results on Compatibility of Parameterization.

In Table 2 of the main paper, we reported an average performance over the unseen test network architecture such as AlexNet, VGG11, and ResNet18 for evaluating the cross-architecture generalization. Herein, we provide detailed performance for each test network architecture. Table 6 repeatedly shows the significant performance improvement of FreD in terms of cross-architecture generalization. These experimental results validate the effectiveness of frequency domain-based parameterization on both the dataset distillation objective and unseen test architectures.

Table 6: Test accuracies (%) on CIFAR-10 under various dataset distillation loss and cross-architecture. We distill the synthetic dataset by using ConvNet.

| | | DC | | | DM | | | TM | | |
| --- | --- | --- | --- | --- | --- | --- | --- | --- | --- | --- |
| | IPC | 2 | 11 | 51 | 2 | 11 | 51 | 2 | 11 | 51 |
| | #Params | 61.44k | 337.92k | 1566.72k | 61.44k | 337.92k | 1566.72k | 61.44k | 337.92k | 1566.72k |
| AlexNet | Vanilla | 20.0 ±1.3 | 22.4 ±1.4 | 29.5 ±0.9 | 20.7 ±3.6 | 37.0 ±0.9 | 49.1 ±0.9 | 26.1 ±1.0 | 36.0 ±1.5 | 49.2 ±1.3 |
| | w/ IDC | 26.8 ±1.8 | 41.5 ±0.5 | 44.2 ±0.7 | 36.4 ±1.1 | 47.7 ±0.6 | 59.2 ±0.7 | 32.5 ±2.2 | 43.7 ±3.0 | 54.9 ±1.1 |
| | w/ HaBa | 22.2 ±1.1 | 33.0 ±0.9 | 33.4 ±1.4 | 32.1 ±0.6 | 44.1 ±0.7 | 53.1 ±0.9 | 43.6 ±1.5 | 49.0 ±3.0 | 60.1 ±1.4 |
| | w/ FreD | 39.8 ±0.4 | 42.4 ±0.6 | 46.4 ±0.5 | 46.4 ±0.7 | 55.7 ±0.5 | 65.7 ±0.5 | 44.1 ±1.3 | 55.9 ±0.8 | 65.9 ±0.8 |
| VGG11 | Vanilla | 28.0 ±0.3 | 35.9 ±0.7 | 38.7 ±0.5 | 22.3 ±0.0 | 41.6 ±0.6 | 55.2 ±0.5 | 38.0 ±1.2 | 50.5 ±1.0 | 61.4 ±0.3 |
| | w/ IDC | 34.3 ±0.7 | 40.0 ±0.5 | 42.4 ±0.8 | 38.2 ±0.6 | 52.8 ±0.5 | 62.2 ±0.3 | 48.2 ±1.2 | 52.1 ±0.7 | 65.2 ±0.6 |
| | w/ HaBa | 29.4 ±0.9 | 37.0 ±0.4 | 41.9 ±0.6 | 26.9 ±0.6 | 49.4 ±0.4 | 67.5 ±0.4 | 48.3 ±0.5 | 60.5 ±0.6 | 67.5 ±0.4 |
| | w/ FreD | 38.8 ±0.9 | 40.0 ±0.8 | 44.8 ±0.9 | 48.1 ±0.9 | 59.0 ±0.6 | 66.6 ±0.2 | 51.0 ±0.8 | 60.0 ±0.6 | 69.9 ±0.4 |
| ResNet18 | Vanilla | 18.1 ±0.8 | 18.4 ±0.4 | 22.1 ±0.4 | 22.3 ±1.0 | 40.0 ±1.5 | 53.4 ±0.7 | 35.2 ±1.0 | 45.1 ±1.5 | 54.5 ±1.0 |
| | w/ IDC | 24.9 ±0.9 | 24.8 ±0.7 | 34.1 ±0.7 | 37.3 ±1.5 | 50.9 ±0.7 | 62.5 ±0.5 | 46.7 ±0.9 | 50.2 ±0.6 | 64.5 ±1.2 |
| | w/ HaBa | 24.5 ±0.6 | 24.3 ±0.6 | 31.1 ±0.3 | 31.3 ±0.7 | 47.6 ±0.5 | 59.6 ±0.4 | 47.4 ±0.7 | 58.0 ±0.9 | 64.4 ±0.6 |
| | w/ FreD | 33.0 ±1.1 | 29.8 ±0.6 | 37.0 ±0.9 | 49.7 ±0.3 | 57.3 ±1.2 | 62.6 ±1.0 | 53.9 ±0.7 | 64.4 ±0.6 | 71.4 ±0.7 |

## D.5    More Results on Robustness against Corruption.

Our proposed method, FreD, demonstrates substantial robustness against corruption, evidence supported by the findings in Figure 7 and Table 5 of the main paper. In this context, we provide further experimental results: 1) the test accuracies results for CIFAR-10.1, and 2) a detailed breakdown of test accuracies based on different types of corruption in CIFAR-10-C. For detailed results on CIFAR-10-C, we report the performance of the severest level and average across all severity levels. In Figure 1, FreD achieves the best performance with a significant gap over the baseline methods on CIFAR-10.1. Furthermore, Table 7 verifies the superior robustness regardless of corruption type.

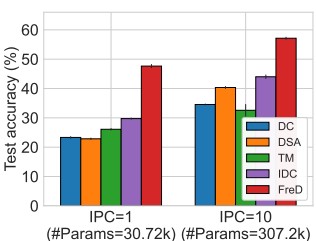

Figure 1: Test accuracies (%) on CIFAR-10.1

Table 7: Test accuracies (%) on CIFAR-10-C with IPC=1 (#Params=30.72k).

(a) Severity level 5 (most severest)

| | Gauss. | Shot | Impul. | Defoc. | Glass | Motion | Zoom | Snow | Frost | Fog | Brit. | Contr | Elastic | Pixel | JPEG | Avg. |
| --- | --- | --- | --- | --- | --- | --- | --- | --- | --- | --- | --- | --- | --- | --- | --- | --- |
| DC | 28.0 | 28.1 | 27.5 | 28.4 | 28.1 | 27.9 | 27.4 | 27.7 | 27.9 | 19.3 | 28.1 | 28.2 | 28.1 | 28.5 | 28.5 | 27.4 |
| DSA | 27.5 | 27.5 | 27.0 | 27.9 | 27.5 | 27.5 | 26.7 | 27.2 | 27.7 | 18.8 | 27.3 | 28.9 | 27.9 | 28.1 | 27.9 | 27.0 |
| TM | 30.2 | 30.4 | 28.6 | 29.0 | 28.0 | 28.2 | 28.6 | 30.4 | 29.6 | 23.0 | 32.5 | 31.4 | 29.3 | 30.0 | 31.2 | 29.4 |
| IDC | 36.4 | 36.2 | 33.3 | 39.4 | 37.6 | 38.6 | 38.4 | 38.1 | 38.7 | 29.7 | 37.9 | 39.1 | 38.4 | 39.2 | 38.7 | 37.3 |
| FreD | 54.4 | 54.2 | 48.9 | 57.1 | 54.8 | 55.4 | 55.4 | 55.9 | 54.7 | 43.3 | 56.3 | 41.6 | 57.1 | 58.5 | 58.1 | 53.7 |

(b) Average across all severity levels

| | Gauss. | Shot | Impul. | Defoc. | Glass | Motion | Zoom | Snow | Frost | Fog | Brit. | Contr | Elastic | Pixel | JPEG | Avg. |
| --- | --- | --- | --- | --- | --- | --- | --- | --- | --- | --- | --- | --- | --- | --- | --- | --- |
| DC | 28.2 | 28.3 | 28.0 | 28.5 | 28.3 | 28.2 | 27.8 | 28.0 | 28.2 | 24.3 | 28.4 | 28.6 | 28.1 | 28.5 | 28.5 | 28.0 |
| DSA | 27.8 | 27.8 | 27.5 | 28.1 | 27.8 | 27.8 | 27.3 | 27.8 | 27.8 | 23.7 | 27.9 | 28.8 | 27.6 | 28.2 | 28.0 | 27.6 |
| TM | 30.8 | 31.1 | 29.9 | 30.5 | 29.0 | 29.5 | 29.6 | 31.0 | 30.5 | 28.0 | 32.3 | 32.4 | 29.5 | 31.1 | 31.5 | 30.4 |
| IDC | 37.4 | 37.8 | 36.3 | 39.7 | 38.2 | 39.0 | 39.0 | 38.9 | 38.6 | 35.7 | 39.3 | 40.4 | 38.5 | 39.4 | 39.1 | 38.5 |
| FreD | 56.7 | 57.3 | 54.4 | 58.9 | 56.4 | 57.2 | 57.3 | 58.0 | 56.5 | 53.6 | 59.2 | 53.5 | 57.2 | 59.6 | 58.9 | 57.0 |

### D.6 Performance Comparison with Memory Addressing

Memory addressing (MA) [6] is a new parameterization method to create a common representation by encapsulating the features shared among different classes into a set of bases. The reported performances of [6] show mixed results under various settings. However, the performance of [6] is not solely due to the MA but also includes the effects of other components. For a fair comparison between MA and FreD, we standardized the distillation loss and evaluated their performances.

Figure 2a shows that MA and FreD exhibit competitive performances on CIFAR-10 with each other under the implemented settings of DM and TM losses. We further assessed the robustness of each approach by evaluating the transferability of the synthetic datasets against diverse distribution shifts. Figure 2b represents mixed result performance on CIFAR-10.1. As a result in Figure 2c, FreD particularly shows better performances than MA in most corrupted versions of datasets. Furthermore, it should be noted that FreD achieves higher performance than MA when the severity level becomes higher. We conjecture that because FreD selects informative dimensions in the frequency domain, it has good robustness to corruptions that typically occur in the high-frequency domain. In terms of computational time, MA requires about nearly three times more than FreD. Please refer to Section E.4 for the detailed discussion.

While both methods are distinct approaches, the implementation of MA in the spatial domain allows a further transformation to the frequency domain. This enables the orthogonal application of MA and FreD. One possible combination is to define the bases of MA in the frequency domain and select the informative dimensions. It allows more flexible parameterization. We leave it as future work.

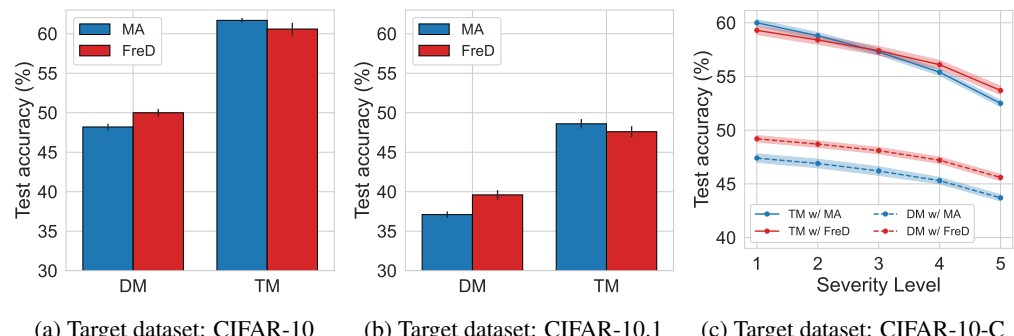

(a) Target dataset: CIFAR-10    (b) Target dataset: CIFAR-10.1    (c) Target dataset: CIFAR-10-C

Figure 2: Performance comparison of MA and FreD on each target dataset (Source dataset: CIFAR-10). Note that higher level indicates higher corruption.

### D.7 Compatibility with BPTT

Back-propagation through time (BPTT) is another optimization framework that effectively solves the bi-level optimization problem. [6] suggests BPTT to train the synthetic dataset in dataset distillation. FreD is a new type of parameterization framework for dataset distillation, while BPTT is introduced as a new optimization framework for dataset distillation. Hence, they can be utilized orthogonally. To verify the efficiency of FreD, we conduct an experiment on models that combine the BPTT framework with FreD.

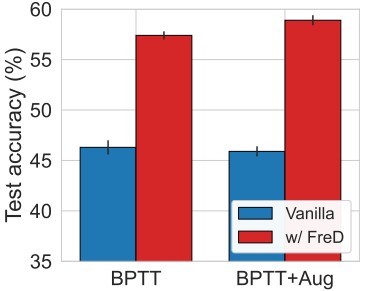

Figure 3: Study of FreD on BPTT

Figure 3 shows the performance of the model with FreD in the BPTT framework on CIFAR-10 under IPC=1 (#Params=30.72k). As mentioned in [6], we considered two variants of BPTT framework with and without augmentation. We reduced the number of training iterations for each baseline and FreD from 50,000 to 5,000.

As a result, BPTT with FreD outperformed BPTT without FreD under BPTT framework regardless of whether or not the augmentation was used. Furthermore, even when compared to the performance of

BPTT with full iteration training reported in the original paper ($49. \pm 0.6$), BPTT w/ FreD achieved higher performance ($57.4 \pm 0.4$). It indicates that FreD is an efficient methodology that can also be applied in the BPTT framework.

## D.8  Additional Ablation Study on Frequency Transform

We basically utilized three frequency transforms: DCT, DFT, and DWT. We especially want to highlight the energy compaction property of DCT, where most of the signal information tends to be concentrated in a few low-frequency components (Please refer to Figure 2 in [1] and Figure 1 in [40]). This characteristic aligns well with the motivation of FreD, and Figure 8b of the main paper demonstrates that DCT is the best choice among the possible options.

To further analyze the effect of frequency transforms on FreD, we have conducted various experiments. Figure 4 presents the results as follows:

- Across all settings, we observe improved performances of FreD than the baseline regardless of the type of frequency transform employed.
- DCT outperforms DFT and DWT in most cases, highlighting the effective exploitation of DCT's energy compaction property within the FreD framework.
- DFT exhibits relatively lower performance in comparison to DCT and DWT. This discrepancy is attributed to the complex-valued nature of DFT. Unlike DCT and DWT, which operate in real space, DFT requires additional resources to represent a single instance due to its complex space. As a result, the quantity of synthetic instances that can be generated within an identical budget is reduced by half than others, leading to lower performance.

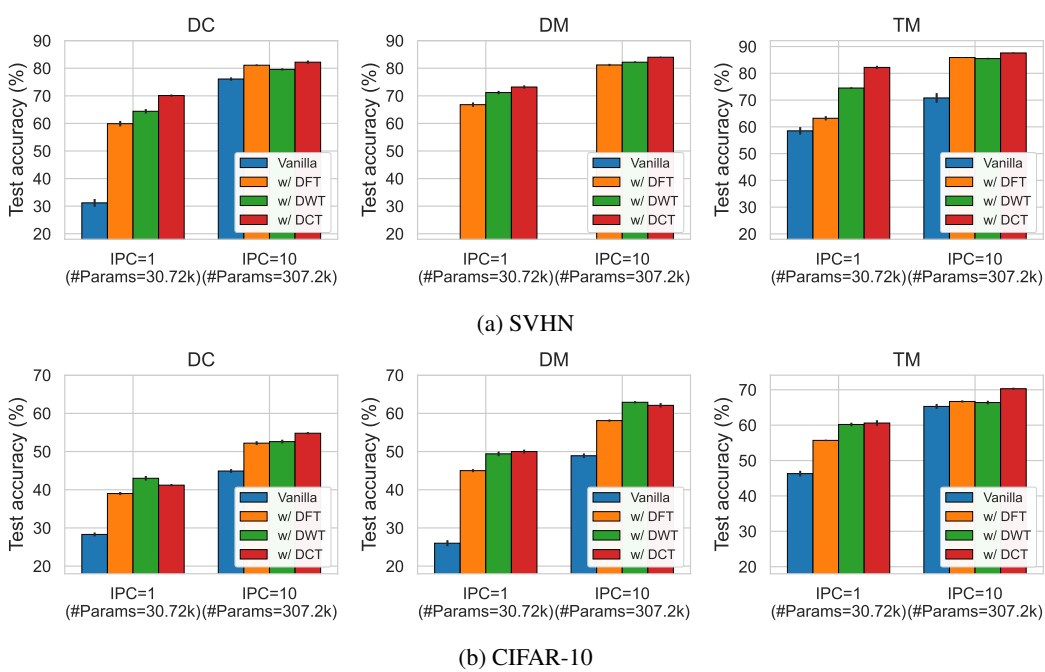

Figure 4: Ablation study on the frequency transform. Note that DM does not provide the test accuracies on SVHN in the original paper.

## D.9  Performance Comparison with Post-downsampling

As mentioned by IDC, the most basic methodology for dataset distillation is to generate the large-cardinality $S$ and compress them with post-processing. In the previous study, the comparison was conducted only in the spatial domain, but this paper extends it to consider post-processing in the frequency domain. Post-processing is the compression of vanilla in each domain. Table 8 presents the results as follows:

- Post-downsampling in both domains achieves lower performance than DM since they compress the trained synthetic dataset. These results indicate the inevitable information loss. While post-downsampling shows information loss, frequency domain-based downsampling achieves higher performance than spatial domain. It demonstrates the frequency domain stores task-relevant information more effectively than spatial domain.

- End-to-end methods achieve higher performance than post-downsampling methods. Among them, the frequency domain-based method (FreD) achieves higher performance than the spatial domain-based method (IDC).

- FreD shows a higher cross-architecture generalization despite spending a quarter of the budget of the vanilla model.

Table 8: Test accuracies (%) comparison under various test network architecture on CIFAR-10. We utilize DM for the dataset distillation loss and ConvNet for the training architecture.

| Decoded instances per class | #Params | Model | | ConvNet | AlexNet | VGG11 | ResNet18 |
|---|---|---|---|---|---|---|---|
| 40 | 1228.8k | DM | | **61.2** ±0.4 | 48.8 ±0.5 | 53.9 ±0.5 | 52.1 ±0.5 |
| | 307.2k | Post-downsampling | Spatial | 56.7 ±0.5 | 44.6 ±0.8 | 49.9 ±0.6 | 49.5 ±0.6 |
| | | | Frequency | 59.3 ±0.4 | 47.4 ±0.5 | 52.5 ±0.5 | 51.2 ±0.6 |
| | | End-to-End | IDC | 59.6 ±0.5 | 47.6 ±0.7 | 52.2 ±0.6 | 50.8 ±0.5 |
| | | | FreD | 60.5 ±0.3 | **50.9** ±0.4 | **54.8** ±0.3 | **53.1** ±0.9 |

### D.10 More Visualization of Binary Mask and Transformed Images

We provide the binary mask and transformed images from our proposed method on various datasets: SVHN (see Figure 9), CIFAR-10 (see Figure 10), CIFAR-100 (see Figure 11a), Tiny-ImageNet (see Figure 11b), and ImageNet-Subset (see Figure 12 and 14). For CIFAR-100 and Tiny-ImageNet, we visualize the first 10 classes. For a better layout, we plot these visualizations at the end of the paper. Through these results, we can observe that the constructed synthetic dataset by FreD contains both intra-class diversity and inter-class discriminative features, regardless of the image resolution.

For 3D MNIST experiments, we provide Figure 8, which displays the original image and a set of trained synthetic data through each distillation method. To enable visualization of the $16 \times 16 \times 16$ dimension point cloud, we sliced each instance's depth dimension into 16 images and displayed them separately. The image located at the top left represents the frontmost view, while the image at the bottom right corresponds to the rearmost view. From Figure 8, FreD effectively captures the class-discriminative information that class 0 should possess. It indicates that the proposed frequency-based dataset distillation framework is applicable to higher-dimensional data than two-dimensional data. Furthermore, compared to the DM and IDC, the synthesized instance by FreD shows more clearer boundary in dimensions $6 \sim 11$ which is the key class-discriminative information of 0. This result demonstrates that the selection of informative dimensions in the frequency domain is effective. In the revised paper, we will add the visualization of the 3D MNIST cloud synthesized by each method.

## Appendix E  Additional Discussions

### E.1 Comparison between FreD and PCA-based Transform

PCA-based transform, which sets the principal components of the given dataset as new axes, can ideally preserve the sum of explained variance ratio by selecting top-$k$ principal components as a subset of new dimensions. However, there are some evidence for the claim that PCA cannot be practically utilized as a method of dataset distillation.

First, PCA-based transform requires an additional budget to store the transform matrix. PCA-based transform utilizes top-$k$ principal components as new axes of the introduced domain. As these axes are composed of a weighted sum of each dimension value, and therefore, it is not possible to implement a feature like FreD, which selects a

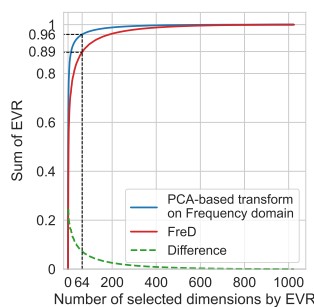

Figure 5: PCA vs FreD.

subset of dimensions from the overall dimensions of the domain. Therefore, the transform matrix created for projection is a $d \times k$ dimensional matrix consisting of the top-$k$ principal components. This matrix needs to be stored separately from the condensed dataset $S$, as it represents a distinct component for transformation, which means an additional budget is needed. Unlike PCA-based transforms, in the frequency domain transform, once you choose a specific frequency transform, the corresponding transform function and inverse transform function remain fixed. Therefore, there is no need to manage these functions separately with an additional budget.

Secondly, the commonly used linear PCA fails to capture the correlations present in the spatial domain of the data (e.g., correlations between adjacent pixels in an image). Although there are spatial principal component analysis [34] methods specifically designed for spatial domains, such methodologies utilize spatial kernel matrix to model the correlation information between adjacent pixels, which could introduce the possibility of information loss. In this subsection, we refer to information loss specifically to the loss of information that occurs during the utilization of a spatial kernel or converting kernel-extracted information into linear features. In other words, it is challenging to accurately determine the principal components for a given dataset during the implementation, making it difficult to use PCA transforms.

Having said that, we conducted the comparison between 1) the sum of explained variance ratio (EVR) obtained through principal component analysis using a dataset transformed into the frequency domain and 2) the sum of EVR by using FreD, which is based on dimension selection in the frequency domain. The sum of EVR based on eigenvectors is maximum in terms of other comparable baselines. However, it should be noted that even if PCA is performed based on the frequency domain, the constraint of storing the projection matrix in the memory budget still remains. Figure 5 illustrates the Cumulative EVR based on the different number of selected dimensions for each method. In our whole experiments, the smallest dimension selection was 64 dimensions. Based on this dimension selection, the Cumulative EVR of FreD, compared to the sum of the EVR of the top-64 eigenvectors in PCA, differs by only around 7%. Furthermore, when more dimensions are selected, this difference becomes smaller. In this regard, FreD can be considered an efficient methodology that sacrifices slightly in EVR while not requiring an additional memory budget for an additional transform matrix.

## E.2 More Visualization of $\|\nabla_S \mathcal{L}_{DD}(S, D)\|$ in Frequency Domain

The main idea of this paper is to compress spatial domain information into fewer frequency dimensions. To verify our idea, we investigate the magnitude of the gradient for $\mathcal{L}_{DD}(S, D)$ i.e. $\|\nabla_S \mathcal{L}_{DD}(S, D)\|$. Specifically, we visualize the magnitude of the gradient of DM distillation loss in Figure 2b of the main paper. To demonstrate that our observation is not confined to a specific distillation loss, we provide the magnitude of the gradient in the frequency domain across different distillation losses, which are DC and TM in Figure 6. It shows that the concentration of gradient exhibits a consistent pattern regardless of the type of distillation loss.

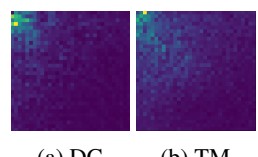

(a) DC      (b) TM

Figure 6: Visualization of $\|\nabla_S \mathcal{L}_{DD}(S, D)\|$

## E.3 Discussion on Budget Allocation

In this section, we will delve further into how the frequency domain-based dimension subset selection by FreD leads to a reduction in the actual budget. Let $x \in \mathbb{R}^{d_1 \times d_2}$ as a data instance in the spatial domain, whose dimension size is $d = d_1 \times d_2$. If each element of $x$ is a 32-bit float, each image would occupy $32 \times d$ bits in memory. Let's assume the same instance is transformed into the frequency domain with the same dimension size, and we only utilize $k$ dimensions in that domain. In that case, the budget required to represent the values would decrease to $32 \times k$ bits. However, in addition to simply storing the values on selected dimensions, we also need to store information about the positions where each value is located. One advantage of FreD is that instead of having separate masks for each instance, it has separate masks for each class. It means that we only need to store the position of the dimension being passed through, once per class. Therefore, we can prevent budget waste by storing the indices of the selected dimension $\tilde{M}$ and the frequency coefficient value of that

dimension $\tilde{f}$, rather than storing the entire frequency representation $f$:

$$f = \mathcal{F}(x) = \begin{pmatrix} 0.2335 & 0.0000 & 0.1246 \\ 0.1243 & 1.0442 & 0.0000 \\ 0 & 0.0000 & 0.0000 \end{pmatrix} \iff \begin{cases} \tilde{f} = [0.2335, 0.1246, 0.1243, 1.0442] \\ \tilde{M} = [0, 2, 3, 4] \end{cases}$$

It should be noted that the masking list $\tilde{M}$, which contains the dimension indices, only needs to store integers. Additionally, since only one masking list per class is required, it reaches a level that can be ignored in terms of the budget. Therefore, the total required budget becomes $32 \times k$ bits. With the frequency information stored in this manner, it becomes possible to reassemble it into a tensor for future use without any information loss.

### E.4    Algorithm Complexity

As a complexity analysis, we consider the computation time for single image retrieval based on FreD and other parameterization methods. For sized 2D image in a spatial domain, IDC utilizes a resizing method as a parameterization, so its complexity is $\mathcal{O}(HW)$ where $H$ is the height and $W$ is the width of the image. HaBa, which requires an additional network for computation, inherits the complexity, $\mathcal{O}(HWF^2)$, where $F$ is the filter size of the convolution neural network. MA operates by performing matrix multiplication between the matrix and downsampled bases. Given that $K$ represents the number of bases and s is the downsampling scale, MA's complexity is $\mathcal{O}(HW \times \frac{K}{s^2})$. FreD, when used with DCT, involves two operations: masking and inverse frequency transform. The complexities of these steps are $\mathcal{O}(HW)$ and $\mathcal{O}(HW \log W)$ respectively. Thus, the total complexity of FreD is $\mathcal{O}(HW \log W)$.

Based on the above complexity, Figure 7 shows the empirical wall-clock time of single-image retrieval for each method. As a result, we show that FreD inherits the second-best single-image retrieval complexity. Although IDC is most efficient in time complexity, we empirically demonstrated that the parameterization based on IDC falls behind FreD in terms of performance in most settings. Furthermore, we want to note that HaBa and GLaD, which utilize the parameterized transform, show extremely high computation cost in terms of single-image retrieval.

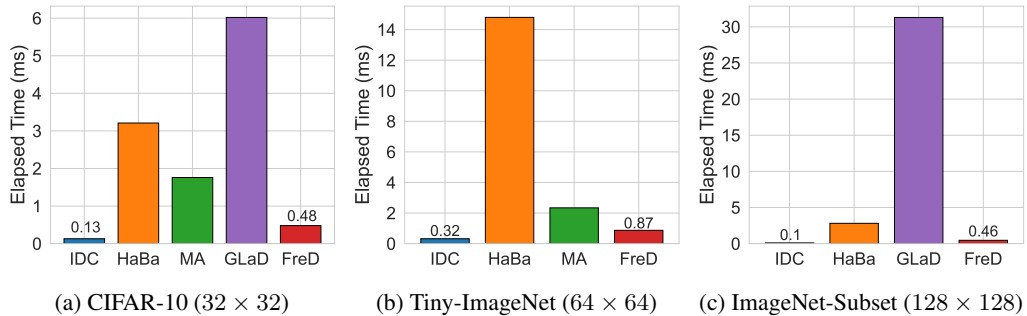

(a) CIFAR-10 ($32 \times 32$)    (b) Tiny-ImageNet ($64 \times 64$)    (c) ImageNet-Subset ($128 \times 128$)

Figure 7: Wall-clock time of single image retrieval for each method. "ms" denotes the millisecond.

### E.5    Influence of Spectral Bias on Frequency Domain-based Approach

Spectral bias refers to the phenomenon that neural networks are prone to prioritize learning the low-frequency components over relatively higher-frequency components [26, 38, 8]. For FreD, even though we employed a mask based on the explained variance ratio (EVR), which is not a simple low-frequency filter: the low-frequency components in the frequency domain were predominantly selected in most experimental scenarios. It should be noted that our EVR does not enforce keeping the low-frequency components, unlike the neural network's spectral bias; what EVR only enforces is keeping the components with a higher explanation ratio on the image feature.

However, this characteristic can become a risk to performance if the task-specific information of data is mostly found in the high-frequency components. These cases include 1) medical imaging on fine details like tumors [21] and 2) digital watermarking [22]. In such cases, there may be a requirement for new masking that allows FreD to capture important high-frequency components. The masking

strategy of FreD can be flexibly operated, and depending on the characteristics of the given dataset and task, it can readily employ other strategies as needed.

### E.6 Impact of Linear Bijectivity Assumption of $\mathcal{F}$

**Impact of linear bijectivity** For the theory presented in Section 3.3 of the main paper to hold, the function $\mathcal{F}$ must be linearly bijective. Note that FreD could utilize any kind of frequency transforms, such as DCT, DFT, and DWT. Since these transforms are all linearly bijective in theory, the proposed proposition could be applicable without any limitation.

Having said that, we elaborate on the potential issues that might emerge, when integrating not linearly bijective transforms into the framework of FreD, as follows:

**When $\mathcal{F}$ is not bijective.** If $\mathcal{F}$ is not bijective, then its inverse $\mathcal{F}^{-1}$ does not exist. This absence makes the process of transforming to another domain and then restoring back to the original domain infeasible. It potentially results in information loss that interferes accurate reconstruction of the original image. An alternative solution could be separating the transform into distinct encoder and decoder components, which enable a procedure for one-to-one mapping. However, the construction of such components necessitates additional training costs, when $\mathcal{F}$ of FreD does not require any kind of additional training. Furthermore, if the encoding is not one-to-one, information might be lost during the encoding process.

**When $\mathcal{F}$ is not linear but bijective.** FreD employs a subset of dimensions in the frequency domain. The choice is feasible as the linearly bijective transform maintains the EVR in the selected dimensions. Proposition and corollary in Section 3.3 theoretically support this attribute, where these do not apply to nonlinear bijective transforms. For 2D-image processing, nonlinear bijective transforms include 1) Log-Polar Transform, and 2) Radial Basis Function (RBF) transform. Domains from these transforms do not exhibit the concentration of the original dataset's variance on specific dimensions.

## Appendix F  Broader Impact

How to parameterize $S$ for dataset distillation is highly versatile as it enhances efficiency and performance by determining the form of the data optimized and stored, regardless of the form of the objective of dataset distillation. Furthermore, in contrast to previous research that was solely conducted in the spatial domain, the exploration of the frequency domain introduces a new perspective in interpreting datasets. In our study, the analyzed dataset consists of pure images without any injected noise. However, real-world datasets can often contain unintended adversarial noise or other types of noise during the processing stages. Analyzing datasets based on the frequency domain enables the detection of noise that may not be visually apparent to the human eye. Moreover, by separately treating specific frequency information that is susceptible to noise, it becomes possible to extend the research to areas such as noisy-filtered dataset distillation.

## Appendix G  Limitation

**Efficacy Differences Depending on Applied Domain.** The applicability and efficacy of FreD's frequency transform are demonstrated specifically within the spatial domain of 2D/3D images. Among the data domains commonly used in machine learning frameworks, natural language domain would likely be challenging to connect with the frequency domain directly. While Fourier Transform can still be applied to text after it's been converted into a numerical format, such as a time series, this conversion is non-trivial and the resulting frequency domain representation may not be as intuitively meaningful. Thus, for certain domains, the effectiveness of frequency transform may not be as substantial as it is for 2D/3D images.

However, in multi-modal tasks, major components like audio signals and video data naturally align with the frequency domain. Tools such as the 1D Fourier transform for audio signals and 3D Fourier transform for video data already exist to process these types directly. Excluding a few specific domains, FreD would be a framework that can be applied across a broader range of domains.

**Performances Highly Dependent on Masking Strategy.** FreD's frequency-based parameterization is motivated by the fact that the spatial domain information of a 2D image can be concentrated in specific components of the transformed frequency domain. The EVR based masking selects important dimensions from the frequency domain, consistently showing strong performances across the various experiments conducted in this study.

Having said that, Figure 8a of the main paper demonstrates that there could be significant performance disparities depending on the masking strategy employed. Therefore, there could be substantial issues if the chosen masking strategy fails to select the important dimensions accurately. For certain datasets or tasks, essential task-specific information might be contained in the high-frequency region. These cases could include 1) medical imaging on fine details like tumors [21] and 2) digital watermarking [28]. This highlights a limitation that EVR masking may not be suitable for all data and tasks.

It should be noted that the masking strategy of FreD can be flexibly operated, and depending on the characteristics of the given dataset and task, it is not restricted to using an EVR-based mask and can readily employ other strategies as needed.

One potential solution to identify task-related frequency components is to utilize the gradient of the given task loss. If specific frequency components have a substantial gradient distribution, it indicates that the component greatly influences the task. By substituting with gradient-based masking, we could address the potential limitations that EVR masks might have.

## Table 9: List of hyper-parameters.

### (a) Gradient matching (DC)

| Dataset | #Params | Synthetic batch size | Learning rate (Frequency) | Selected dimension per channel | Increment of instances |
|---|---|---|---|---|---|
| CIFAR-10 | 61.44k (IPC=2) | - | $10^3$ | 32 | ×32 |
| | 337.92k (IPC=11) | - | $10^3$ | 128 | ×8 |
| | 1566.72k (IPC=51) | 256 | $10^2$ | 256 | ×4 |
| LSUN | 491.52k (IPC=1) | 80 | $10^5$ | 128 | ×128 |
| ImageNet-Subset (128 × 128) | 491.52k (IPC=1) | - | $10^5$ | 2048 | ×8 |
| ImageNet-Subset (256 × 256) | 1966.08k (IPC=1) | - | $10^6$ | 8192 | ×8 |

### (b) Feature matching (DM)

| Dataset | #Params | Synthetic batch size | Learning rate (Frequency) | Selected dimension per channel | Increment of instances |
|---|---|---|---|---|---|
| CIFAR-10 | 61.44k (IPC=2) | - | $10^6$ | 64 | ×16 |
| | 337.92k (IPC=11) | - | $10^5$ | 128 | ×8 |
| | 1566.72k (IPC=51) | - | $10^5$ | 256 | ×4 |
| LSUN | 491.52k (IPC=1) | 40 | $10^5$ | 256 | ×64 |
| ImageNet-Subset (128 × 128) | 491.52k (IPC=1) | - | $10^6$ | 2048 | ×8 |
| 3D MNIST | 40.96k (IPC=1) | - | $10^6$ | 512 | ×8 |
| | 409.6k (IPC=10) | - | $10^6$ | 1024 | ×4 |
| | 2048k (IPC=50) | - | $10^6$ | 1024 | ×4 |

### (c) Trajectory matching (TM)

| Dataset | #Params | Synthetic steps | Expert epochs | Max start epoch | Synthetic batch size | Learning rate (Frequency) | Learning rate (Step size) | Learning rate (Teacher) | Selected dimension per channel | Increment of instances |
|---|---|---|---|---|---|---|---|---|---|---|
| MNIST | 7.84k (IPC=1) | 50 | 2 | 5 | - | $10^6$ | $10^{-7}$ | $10^{-2}$ | 49 | ×16 |
| | 78.4k (IPC=10) | 30 | 2 | 15 | - | $10^5$ | $10^{-5}$ | $10^{-2}$ | 392 | ×2 |
| Fashion MNIST | 7.84k (IPC=1) | 50 | 2 | 5 | - | $10^6$ | $10^{-7}$ | $10^{-2}$ | 49 | ×16 |
| | 78.4k (IPC=10) | 60 | 2 | 15 | - | $10^5$ | $10^{-5}$ | $10^{-2}$ | 196 | ×4 |
| SVHN | 30.72k (IPC=1) | 50 | 2 | 5 | - | $10^7$ | $10^{-7}$ | $10^{-2}$ | 64 | ×16 |
| | 307.2k (IPC=10) | 30 | 2 | 15 | - | $10^7$ | $10^{-5}$ | $10^{-2}$ | 128 | ×8 |
| | 1536k (IPC=50) | 40 | 2 | 40 | 500 | $10^7$ | $10^{-5}$ | $10^{-3}$ | 256 | ×4 |
| CIFAR-10 | 30.72k (IPC=1) | 50 | 2 | 5 | - | $10^8$ | $10^{-7}$ | $10^{-2}$ | 64 | ×16 |
| | 61.44k (IPC=2) | 50 | 2 | 5 | 160 | $10^8$ | $10^{-7}$ | $10^{-2}$ | 64 | ×16 |
| | 307.2k (IPC=10) | 40 | 2 | 15 | 320 | $10^7$ | $10^{-5}$ | $10^{-2}$ | 160 | ×6.4 |
| | 337.92k (IPC=11) | 40 | 2 | 15 | 320 | $10^7$ | $10^{-5}$ | $10^{-2}$ | 176 | ×5.82 |
| | 1536k (IPC=50) | 30 | 2 | 40 | 500 | $10^7$ | $10^{-5}$ | $10^{-3}$ | 256 | ×4 |
| | 1566.72k (IPC=51) | 30 | 2 | 40 | 510 | $10^7$ | $10^{-5}$ | $10^{-3}$ | 256 | ×4 |
| CIFAR-100 | 30.72k (IPC=1) | 50 | 2 | 15 | - | $10^8$ | $10^{-5}$ | $10^{-2}$ | 128 | ×8 |
| | 307.2k (IPC=10) | 20 | 2 | 40 | 2048 | $5 \times 10^6$ | $10^{-5}$ | $10^{-2}$ | 400 | ×2.56 |
| | 1536k (IPC=50) | 80 | 2 | 40 | 256 | $5 \times 10^6$ | $10^{-5}$ | $10^{-2}$ | 400 | ×2.56 |
| Tiny-ImageNet | 2457.6k (IPC=1) | 30 | 2 | 30 | 400 | $10^9$ | $10^{-4}$ | $10^{-2}$ | 512 | ×8 |
| | 24576k (IPC=10) | 40 | 2 | 40 | 300 | $10^9$ | $10^{-4}$ | $10^{-2}$ | 3840 | ×3.2 |
| | 122880k (IPC=50) | 40 | 2 | 40 | 250 | $10^9$ | $10^{-4}$ | $10^{-2}$ | 3840 | ×3.2 |
| ImageNet-Subset (128 × 128) | 491.52k (IPC=1) | 20 | 2 | 10 | - | $10^9$ | $10^{-6}$ | $10^{-2}$ | 2048 | ×8 |
| | 983.04k (IPC=2) | 20 | 2 | 10 | 80 | $10^9$ | $10^{-6}$ | $10^{-2}$ | 2048 | ×8 |
| | 4915.2k (IPC=10) | 20 | 2 | 10 | 80 | $10^9$ | $10^{-6}$ | $10^{-2}$ | 4096 | ×4 |

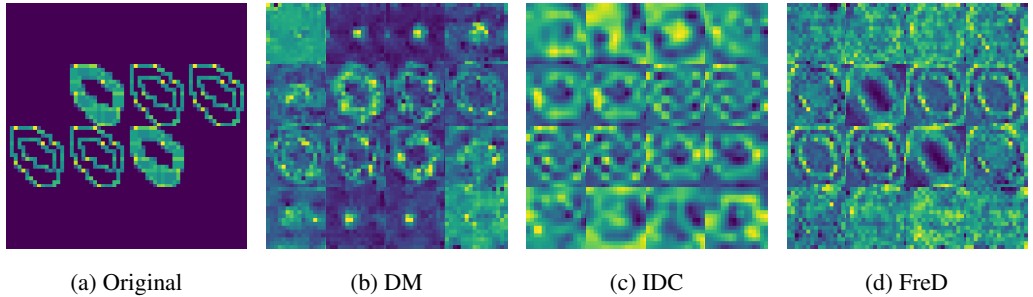

(a) Original       (b) DM       (c) IDC       (d) FreD

Figure 8: The cross-section visualizations of class 0 in 3D MNIST. Each top left image represents the frontmost view, while bottom right image corresponds to the rearmost view.

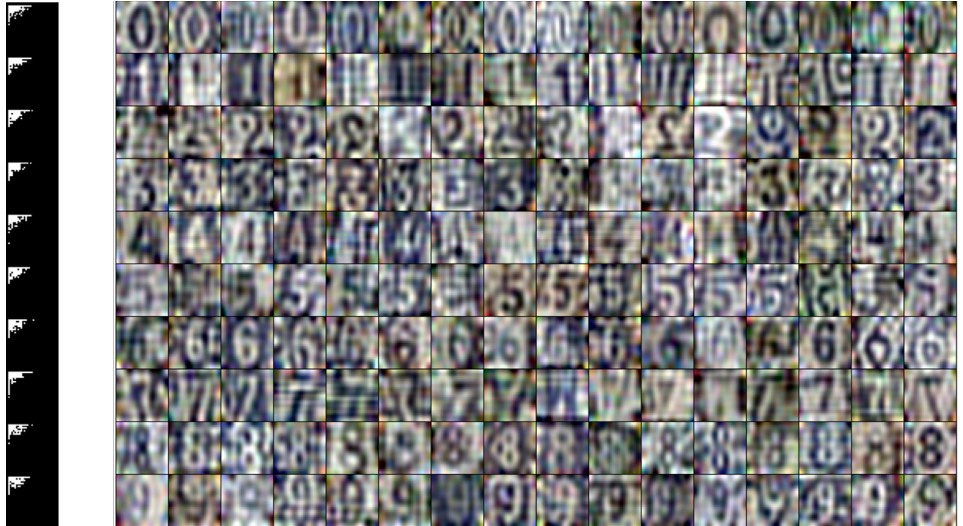

Figure 9: Visualization of the binary mask and the transformed images by FreD on SVHN with IPC=1 (#Params=30.72k). In this setting, FreD constructs 16 images per class under the same budget.

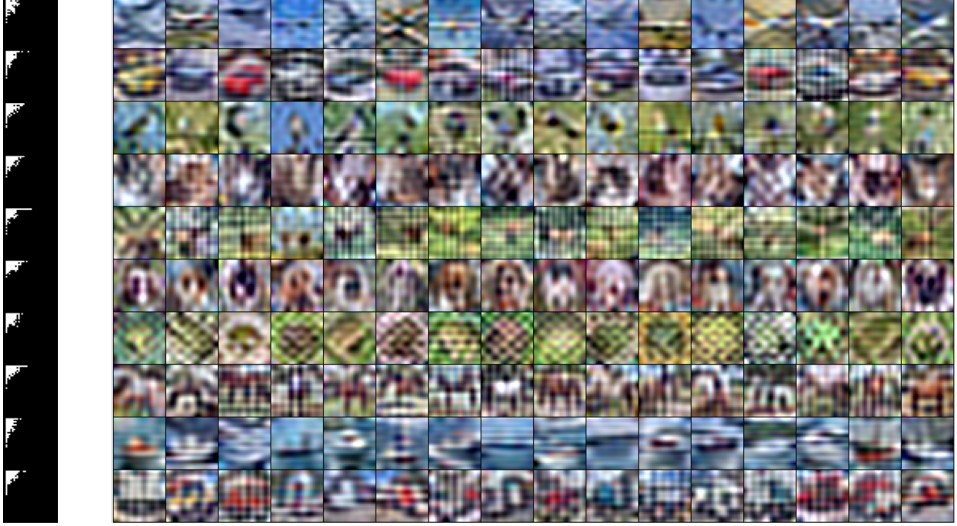

Figure 10: Visualization of the binary mask and the transformed images by FreD on CIFAR-10 with IPC=1 (#Params=30.72k). In this setting, FreD constructs 16 images per class under the same budget.

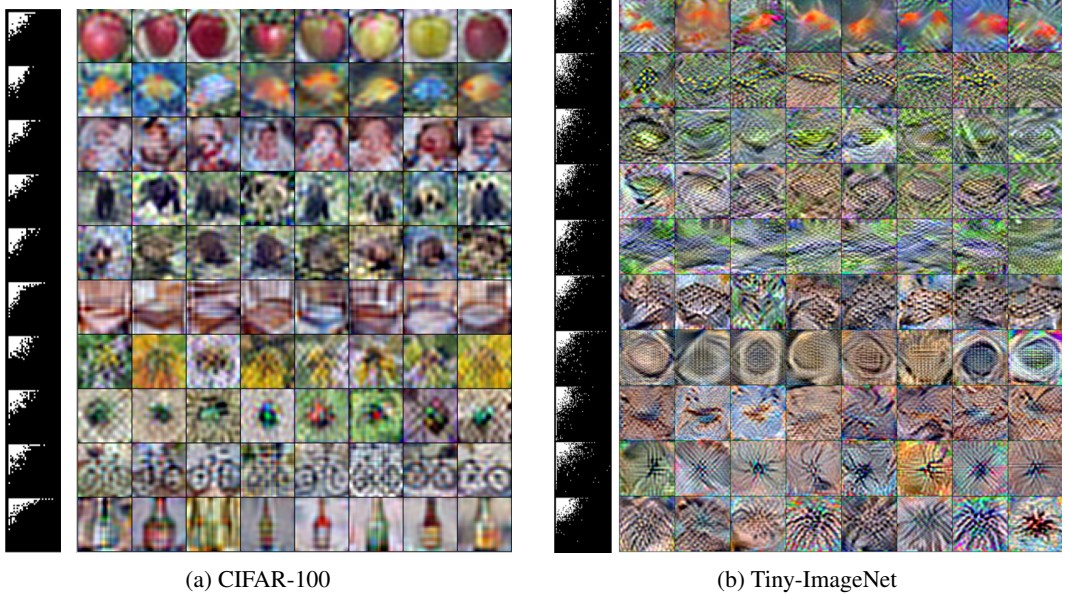

(a) CIFAR-100                                    (b) Tiny-ImageNet

Figure 11: Visualization of the binary mask and the transformed images by FreD on CIFAR-100 with IPC=1 (#Params=30.72k) and Tiny-ImageNet with IPC=1 (#Params=2457.6k). Due to a lack of space, only the first 10 classes were visualized. In both cases, FreD constructs 8 images per class under the same budget.

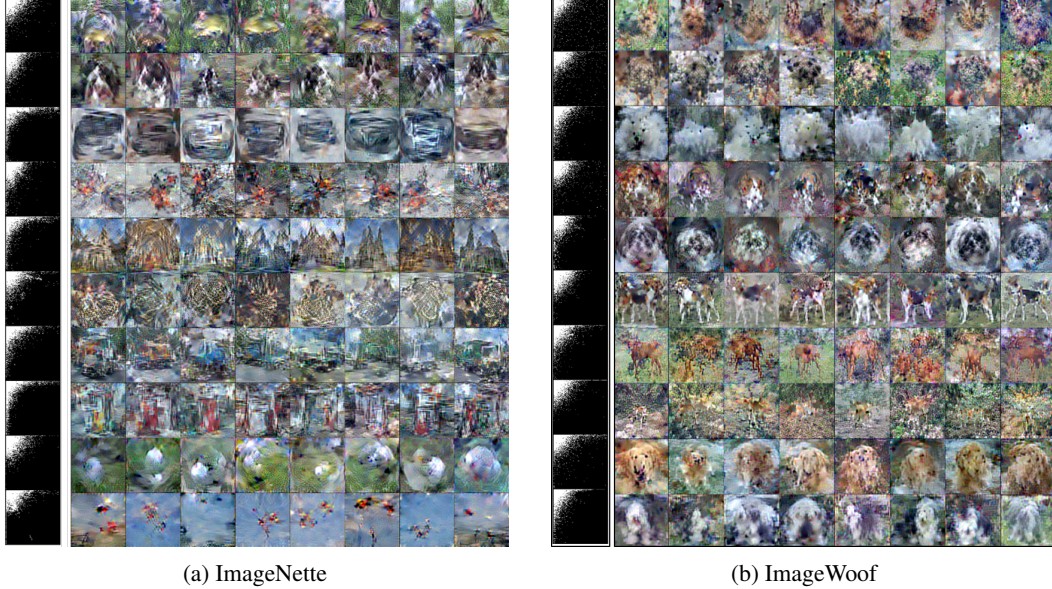

(a) ImageNette                                   (b) ImageWoof

Figure 12: Visualization of the binary mask and the transformed images by FreD on ImageNet-Subset with IPC=1 (#Params=491.52k). In these cases, FreD constructs 8 images per class under the same budget.

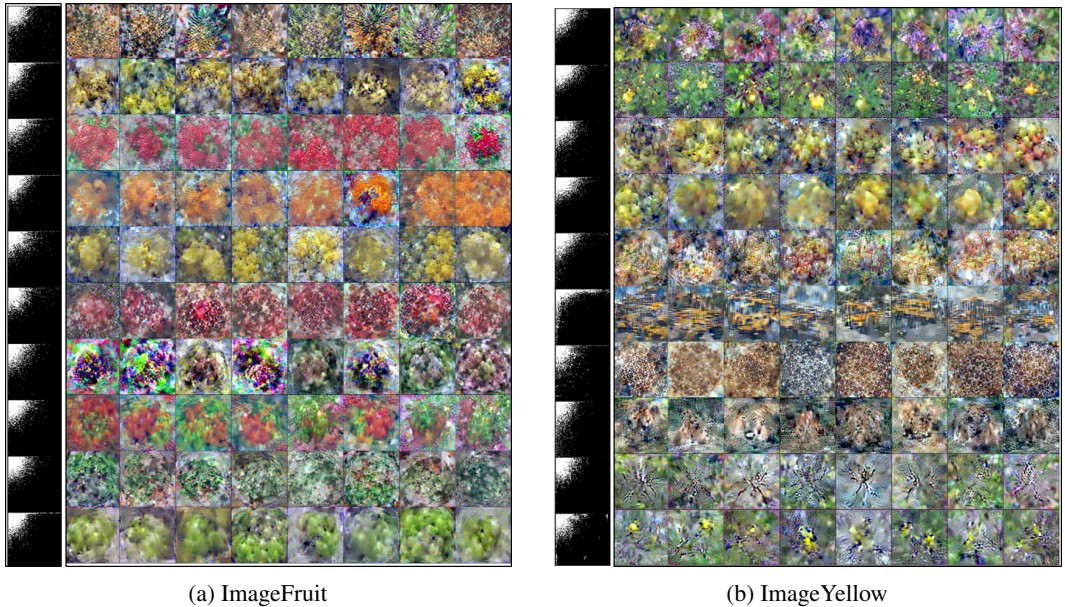

(a) ImageFruit                                      (b) ImageYellow

Figure 13: Visualization of the binary mask and the transformed images by FreD on ImageNet-Subset with IPC=1 (#Params=491.52k). In these cases, FreD constructs 8 images per class under the same budget.

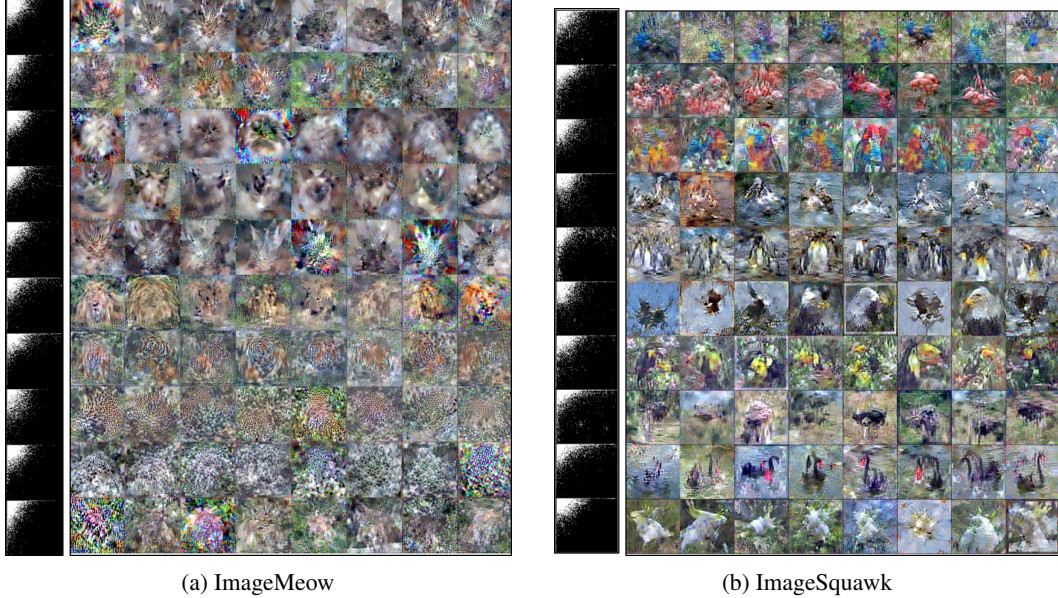

(a) ImageMeow                                      (b) ImageSquawk

Figure 14: Visualization of the binary mask and the transformed images by FreD on ImageNet-Subset with IPC=1 (#Params=491.52k). In these cases, FreD constructs 8 images per class under the same budget.