# OpenReview forum: "Frequency Domain-Based Dataset Distillation"
_NeurIPS.cc/2023/Conference — NeurIPS 2023 poster_

### Official Review · Reviewer_Xfn3 · 2023-07-05

**Soundness:** 3 good
**Presentation:** 3 good
**Contribution:** 3 good
**Rating:** 5
**Confidence:** 5

**Summary:**

This paper proposes a frequency-domain dataset distillation approach that can help to select the most important information in images to help distillation. To do this, the approach starts with a DCT to the input $x$ and then performs loss matching using existing dataset distillation approaches. Experiments demonstrate its advantages over state-of-the-art approaches.

**Strengths:**

1. The idea of using frequency domain information is novel and helpful.
2. The experiments are good and solid.
3. The visualization helps to understand the algorithm and results better.

**Weaknesses:**

1. Experiments are solid, but not enough. I think that dataset distillation is useful for large volume datasets. But experiments do not contain any large datasets. So, the experiments are not useful to help interprete the usefulness of the algorithm.
2. There are no transfer experiments. Since dataset distillation is used to store a proxy of large datasets, and large datasets are often act as the source domain for transfer learning. Therefore, authors should conduct experiments on transfer learning datasets to show the advantage of the proposed approach. Only doing experiments on CIFAR-level datasets are not enough since to be honest, CIFAR is small and does not need to distill.
3. Any complexity analysis? How efficient is the algorithm?

**Questions:**

See weakness

**Limitations:**

See weakness

---

> ### Author Rebuttal · Authors · 2023-08-09
>
> We really appreciate your constructive review on our manuscript.
>
> $\textbf{[W1. A request for experiments on a large volume dataset]}$
>
> We appreciate the valuable comments of the reviewer. The volume of the dataset could increase by two factors: either 1) the dimension of each instance in the dataset grows, or 2) the overall size of the dataset increases.
>
> Firstly, we conducted the experiments on several high-dimensional datasets such as Tiny-ImageNet, ImageNet-Subset, and LSUN. Please refer to the details in the Q2 of the global response. The results suggest that FreD may be more efficient than spatial domain-based methods as the data dimension increases, and this is empirically demonstrated through experiments.
>
> Secondly, previous researchers in the dataset distillation only focus on investigating the effect of dimension. Therefore, we also investigate the effect of the overall size of the dataset. We choose LSUN dataset [1] as the large dataset. To verify the performance during the rebuttal period, the original dataset was randomly sampled with 10,000/25,000 instances per class which results in a total 100,000/250,000, and experiments were conducted after downsizing each instance to a $3\times128\times128$ size. Figure 4 in the rebuttal material provides performances of FreD and other baselines on the LSUN dataset. As a result, FreD achieves the best performance compared to the implemented baselines.
>
> $\textbf{[W2. Investigation on transferability of synthetic datasets]}$
>
> We are truly thankful for the feedback on this aspect. It is highly important to verify the transferability of a synthetic dataset learned through dataset distillation, where previous works did not explore the aspect.
>
> As an evaluation on the transferability of synthetic datasets across different methods, we refer to a test-time adaptation scenario, which is an example case of transfer learning. For an evaluation of the transferability of the synthetic dataset which was trained on CIFAR-10 and ImageNet, we utilized the following target datasets:
>
> * CIFAR-10.1 [2] consists of 2000 new test images which have the same classes as CIFAR-10.
> * CIFAR-10-C and ImageNet-C [3] aim at measuring the robustness of object recognition based on CIFAR-10 and ImageNet, respectively. They have 15 types of corruption and each corruption has five levels with level 5 indicating the most corrupted.
>
> We created ImageNet-Subset-C by selecting data from ImageNet-C that matches the ImageNet-Subset classes. To examine the transferability of dataset distillation methods, our experiment was conducted in three stages:
>
> 1) Distill the synthetic dataset from a source dataset.
> 2) Train a neural network from scratch using the synthetic dataset.
> 3) Test the network on the target dataset.
>
> Figure 3 in the rebuttal material shows the results of transferability on CIFAR-10.1 and CIFAR-10-C. FreD shows superior accuracy over the whole setting. Furthermore, Table 4 (bottom) in the rebuttal material also shows that FreD achieves better performances in ImageNet-Subset-C. These experimental results indicate that FreD is an efficient dataset distillation method in terms of transferability.
>
> To explain the rationale, corruptions that diminish the predictive ability of a machine learning model often occur at the high-frequency components. Adversarial attacks and texture-based corruptions are representative examples [4,5]. Unlike FreD, which can selectively store information about an image's frequency distribution, transforms such as factorization or upsampling are well-known for not preserving frequency-based information well. Consequently, previous methods are likely to suffer a decline in predictive ability on datasets that retain class information while adding adversarial noise. In contrast, FreD demonstrates relatively good robustness against distribution shifts by successfully storing the core frequency components that significantly influence class recognition, regardless of the perturbations applied to individual data instances.
>
> $\textbf{[W3. Complexity and efficiency of each algorithm]}$
>
> As a complexity analysis, we consider the computation time for single image retrieval based on FreD and other parameterization methods. For $H\times W$ sized 2D image in a spatial domain, IDC utilizes a resizing method as a parameterization, so its complexity is $\mathcal{O}(HW)$. HaBa, which requires an additional network for computation, inherits the complexity, $\mathcal{O}(HW{F}^{2})$, where $F$ is the filter size of the convolution neural network. MA [6] operates by performing matrix multiplication between the matrix and downsampled bases. Given that $K$ represents the number of bases and s is the downsampling scale, MA's complexity is $\mathcal{O}(HW \times \frac{K}{s^2})$. FreD, when used with DCT, involves two operations: masking and inverse frequency transform. The complexities of these steps are $\mathcal{O}(HW)$ and $\mathcal{O}(HW\log W)$ respectively. Thus, the total complexity of FreD is $\mathcal{O}(HW\log W)$.
>
> Based on the above complexity, Table 2 in the rebuttal material shows the empirical wall-clock time of single-image retrieval for each method. As a result, we show that FreD inherits the second-best single-image retrieval complexity. Although IDC is most efficient in time complexity, we empirically demonstrated that the parameterization based on IDC falls behind FreD in terms of performance in most settings.
>
> [1] LSUN: Construction of a Large-scale Image Dataset using Deep Learning with Humans in the Loop, arxiv2015
>
> [2] Do CIFAR-10 Classifiers Generalize to CIFAR-10?, ICML 2019
>
> [3] Benchmarking NEURAL NETWORK ROBUSTNESS TO COMMON CORRUPTIONS AND PERTURBATIONS, ICLR 2019
>
> [4] Frequency Domain Model Augmentation for Adversarial Attack, ECCV 2022
>
> [5] PatchAttack: A Black-Box Texture-Based Attack with Reinforcement Learning, ECCV 2020
>
> [6] Remember the Past: Distilling Datasets into Addressable Memories for Neural Networks, NeurIPS2022

---

> > ### Comment · Reviewer_Xfn3 · 2023-08-18
> > **Thanks for your response**
> >
> > I would like to thank the authors for your response and most of my concerns are resolved. I will not change my rating to remain a positive side of this paper.

---

> > > ### Author Response · Authors · 2023-08-18
> > > **Thanks for reviewer Xfn3**
> > >
> > > Thank you for taking the time to review our work and for your positive comments.
> > >
> > > Also, we're glad to hear that most of your concerns have been addressed!
> > >
> > > If you have further uncleared concerns, we are more than eager to provide additional clarification.

---

### Official Review · Reviewer_5mRL · 2023-07-05

**Soundness:** 3 good
**Presentation:** 3 good
**Contribution:** 3 good
**Rating:** 6
**Confidence:** 4

**Summary:**

The paper proposes a novel way (code named FreD) to perform parameterization on dataset distillation tasks. Different from other methods which all operate on the spatial domain, this paper proposes to convert the spatial domain into frequency domain which are more concentrated and then perform distillation on these concentrated frequency components.

Empirical evaluations show that FreD is able to achieve SOTA compared to previous methods.

============

I have read the author's responses. Part of my concerns are addressed such as the questions regarding IPC while some others remain debatable. Overall, I would still keep my original score.

============

**Strengths:**

## originality
- the paper proposes an interesting new way to perform dataset distillation which has not be explored previously to the best of my knowledge

## quality
- the paper shows the effectiveness of the proposed methods through an adequate amount of experiments
- the method is also theoretically proved through proposition 1 and corollary 1.

## clarity
- the paper is well written and easy to follow
- the paper categorizes baseline methods clearly which makes the evaluation results easy to follow
- the increment of decoded instances are also clearly indicated in table 1
- distilled results are visualized and showed in the paper

## significance
To the best of my knowledge, previous parameterization methods mostly perform on the spatial domain. This paper is the first to propose converting spatial domain to frequency domain which concentrates the information better. It could potentially open up a new directions for dataset distillation/condensation methods.

**Weaknesses:**

- The IPCs used in table 1 are 1/10/50, but 2/11/51 in table 2. The motivation for doing so is not clear.
- In Figure 2, does the image used come from the evaluation datasets? If not, it's confusing to use an image that is not related to the  evaluation datasets in this paper.
- The evaluation datasets are all small datasets such as 32 * 32 , in the appendix, only IPC 1 for TinyImageNet is reported. What's the reason for the missing numbers on IPC 10/50? Is it scalability issues or performances issues or something else?
- In [1], IPC 1 and 10 are used to evaluate the performance on ImageNet subsets, but IPC 2 is used in Fred, what's the specific reason to choose a different IPC than previous methods? Can we see some results on IPC 1 and 10 as well because it will be good to see how FreD performs on higher resolutions.

[1] Dataset Distillation by Matching Training Trajectories

**Questions:**

See my comments on weakness. I am willing to raise the score if the above questions are answered so that I can get a better idea of how FreD works beyond what's currently shown in the main paper.

**Limitations:**

- the evaluation in the main paper is only done on datasets with low resolution, thus the scalability is hard to verify
- same with a lot of previous parameterization methods, FreD also seem to work best with smaller IPCs  such as IPC 1(where generating more images can boost the performance a lot). This can be seen on CIFAR-100 IPC 50 that even with 2.56 times more images, the performance is close to methods without parameterization.

---

> ### Author Rebuttal · Authors · 2023-08-09
>
> We really appreciate your constructive review on our manuscript.
>
> $\textbf{[W1. Differences in IPC settings for each table]}$
>
> We apologize for any confusion caused to the reviewer by utilizing different IPC values across each table. In the case of HaBa, which is FreD's main baseline, conducting an experiment at IPC=1 is structurally impossible due to the characteristics of HaBa. HaBa requires the learning of two components during the distillation process: the hallucination networks and the bases. Even though they adopted lightweight networks, the utilized hallucination network occupies the budget corresponding to IPC=1, which disables the learning bases on the setting of IPC=1. This is reflected in HaBa's original paper, where they reported the performance at IPC=2/11/51 due to this limitation. For your reference, in Table 1 of the main paper, HaBa's IPC=1 performance is actually measured with an IPC=2 budget. We have identified that this is not clearly written and will add an explanation in the revised paper.
>
> While FreD and IDC do not have this constraint, we conducted the experiments at IPC=2/11/51 in Table 2-3 in the main paper and Table 2 in the supplementary material to facilitate a fair comparison with HaBa. For your information, you can check the performance of FreD on CIFAR-10, IPC=1/10 case in Table 3 in the rebuttal material. Please see the "w/ DCT" case.
>
> $\textbf{[W2. Specification of the image on Figure 2 in the main paper]}$
>
> We utilized an instance of airplane class in CIFAR-10 for Figure 2 in the main paper. For the convenience of analysis, it has been converted to a gray image. It does not hurt the applicability of FreD, because the transformation to the frequency domain is applied independently for each channel.
>
> $\textbf{[W3. Questions on additional experiments on TinyImageNet and the presence of scalability issues]}$
>
> We report the experiment results on high-dimensional data, such as Tiny-ImageNet ($3\times64\times 64$), ImageNet-Subset ($3\times128\times 128$), and LSUN ($3\times128\times128$) in the supplementary material (Table 1, 2) and rebuttal material (Table 4 and Figure 4). Please check the Q2 of the global response for the details.
>
> In an experiment with the Tiny-ImageNet at a setting of IPC=10, the TM model recorded an accuracy of 23.2%, while FreD showed an improvement with an accuracy of 24.15%. In the IPC=50 case, the TM recorded an accuracy of 28.0%, while FreD recorded 26.4%.
>
> FreD's motivation lies in leveraging select important dimensions of the frequency domain, which can contain much of the spatial domain's information, to utilize the given memory budget more efficiently. This efficiency manifests greater utility when the available memory budget is more limited, and as the reviewer pointed out, FreD demonstrates more substantial performance improvement in most experiments with an IPC=1 setting.
>
> TinyImageNet is originally a dataset with 500 instances per class, and the IPC=50 setting for this dataset could be considered a not-so-drastic reduction.
>
> In situations where such a significant reduction doesn't occur, FreD's motivation may be weakened. Excluding this particular setting, FreD consistently demonstrates superior performance compared to the baseline across evaluations with more diverse datasets that have high resolution and large volumes. Please refer to the supplementary material (Table 1, and 2) and the rebuttal material (Table 4 and Figure 4) for the results.
>
> We conjecture that the FreD has small computation complexity in terms of data retrieval. FreD consists of two operations: EVR masking and inverse frequency transform. Both operations do not require heavy computation costs. We demonstrate that FreD's complexity of single-image retrieval in the response of W3 to Reviewer Xfn3. For empirical evidence, Table 2 in the rebuttal material indicates the empirical wall-clock time of single-image retrieval for parameterization methods. As a result, FreD achieves the second-best single-image retrieval complexity.
>
> $\textbf{[W4. Question on the selection of different IPC than previous methods and request on the additional experiments]}$
>
> Please refer to the response of W1 for the specific reason for choosing IPC=2 for performance comparison. We additionally report the performance of IPC=1/10 on ImageNet-Subset in Table 4 (Top) in the rebuttal material. Due to the time constraint imposed by the rebuttal deadline, we reduced the number of training iterations for each baseline and FreD from 15000 to 3000. It should be noted that the performance of FreD at IPC=1/10 already overwhelms the performance of HaBa at IPC=2/11, respectively. In the revised paper, we will reflect on the final performance based on the ultimate iteration of training.

---

### Official Review · Reviewer_tks8 · 2023-07-05

**Soundness:** 4 excellent
**Presentation:** 4 excellent
**Contribution:** 4 excellent
**Rating:** 8
**Confidence:** 5

**Summary:**

This work introduces Frequency-based Dataset Distillation as a new means of parameterizing the distilled dataset that requires a much smaller memory footprint. By learning the distilled dataset in the frequency space (defined by the discrete cosine transform by default), a binary mask based on the class-wise EVR can be used to select only certain frequency components with minimal performance tradeoff. The memory saved by masking out these components can be allocated towards distilling more total samples.

**Strengths:**

This work is well written and includes an exhaustive suite of experiments showing the effectiveness of the method.

Nearly all questions I had while reading were addressed later on in the paper.



**Weaknesses:**

It is not clear which dataset distillation loss is used in Figure 2. Likewise, it is unclear which distillation loss is used by FreD in Table 1 (although I think it is TM based on info in the appendix).

I would recommend removing IPC from the tables all-together and including the number of parameters instead. For example, instead of IPC = 10 for CIFAR-10, you could have #Params = 30,720 (10x3x32x32). IPC can be misleading when comparing to re-parameterization methods since the IPC simply isn't true anymore.

Would it be possible to see any visualizations for 3D MNIST? I searched through the appendix but only found visualizations for the 2D datasets. I've been very curious to see dataset distillation done on 3D data.



**Questions:**

I really like the paper overall and would be happy to raise my score if the authors address the few weaknesses described above.

---

> ### Author Rebuttal · Authors · 2023-08-09
>
> We really appreciate your constructive review on our manuscript.
>
> $\textbf{[W1. Specifications of the distillation loss utilized in the experiment and figures]}$
>
> Thanks for pointing out our lack of explanation. Figure 2 in the main paper utilized DM as a distillation loss to compute the gradient. To demonstrate that our observation is not confined to a specific loss, Figure 1 in the rebuttal material illustrates the magnitude of the gradient in the frequency domain across different distillation losses, which are DC and TM. It shows that the concentration of gradient exhibits a consistent pattern regardless of the type of distillation loss. In the revised paper, we will clarify the information on the utilized dataset distillation loss in Figure 2 of the main paper.
>
> Regarding Table 1 in the main paper, we acknowledge that the reviewer found the answer from section D.3 in the supplementary material. Yes. Table 1 utilizes TM as a distillation loss. We will add the description of utilized distillation loss in the revised paper.
>
> $\textbf{[W2. Suggestion on representation unit of memory budget]}$
>
> Thank you for your valuable suggestion. We acknowledge that the previous researches have utilized Image-per-Class (IPC) as a measure of memory budget, but we agree that the IPC unit is not suitable for analyzing a parameterization method like FreD. We also want to take into account class-shared parameterization, such as the memory addressing in [1], and instead of writing the number of parameters per class, we want to write the total number of parameters in that setting. Specifically, we will replace all IPC with $NumParams$, which follows the form of $Number \quad of \quad instances \times Parameters \quad of \quad instance$. For example, IPC=1 for CIFAR-10 will be replaced to $NumParams=10\times3072$. We strongly agree with changing the way memory budgets are expressed, but to avoid unnecessary confusion, we're sticking with IPC until the discussion period. After discussion period, we will reflect it in the revised paper.
>
> $\textbf{[W3. Visualizations of 3D-MNIST]}$
>
> As the reviewer has suggested, we provide Figure 2 in the rebuttal material, which displays the original image and a set of trained synthetic data through each distillation method. To enable visualization of the $16\times16\times16$ dimension point cloud, we sliced each instance's depth dimension into 16 images and displayed them separately. The image located at the top left represents the frontmost view, while the image at the bottom right corresponds to the rearmost view. From Figure 2 in the rebuttal material, FreD effectively captures the class-discriminative information that class 0 should possess. It indicates that the proposed frequency-based dataset distillation framework is applicable to higher-dimensional data than two-dimensional data. Furthermore, compared to the DM and IDC, the synthesized instance by FreD shows more clear boundary in dimensions 6~11 which is the key class-discriminative information of 0. This result demonstrates that the selection of informative dimensions in the frequency domain is effective. In the revised paper, we will add the visualization of the 3D-MNIST cloud synthesized by each method.
>
> [1] Remember the Past: Distilling Datasets into Addressable Memories for Neural Networks, NeurIPS2022

---

> > ### Comment · Reviewer_tks8 · 2023-08-14
> > **Response to Rebuttal**
> >
> > Thank you for answering all my questions; I will raise my score to an 8.

---

> > > ### Author Response · Authors · 2023-08-15
> > > **Thanks for reviewer tks8**
> > >
> > > Thanks for your work on the reviews and the feedback.
> > >
> > > Please leave a comment if you have further questions.
> > >
> > > We would be happy to provide additional clarification.
> > >
> > > Again, Thank you so much for your time.

---

### Official Review · Reviewer_ajn2 · 2023-07-09

**Soundness:** 2 fair
**Presentation:** 3 good
**Contribution:** 3 good
**Rating:** 4
**Confidence:** 4

**Summary:**

This paper introduces a new parameterization method for dataset distillation, called FreD, that utilizes the frequency domain to distill a compact, synthetic dataset from a large-sized original one. This approach differs from conventional methods that focus on the spatial domain. It employs frequency-based transforms to optimize the frequency representations of each data instance. By leveraging the concentration of spatial domain information on specific frequency components, FreD is able to select a subset of frequency dimensions for optimization. The paper provides both theoretical and empirical evidence of FreD's efficiency compared to conventional parameterization methods. In addition, it highlights the compatibility of FreD with existing distillation methods and its consistent improvement in performance over different benchmark datasets. This work addresses the challenges posed by big data and it improves upon the current dataset distillation methods by offering an optimized and memory-efficient solution.

**Strengths:**

1. The authors have done a great job of describing their proposed method, FreD. They offer explicit step-by-step explanations of how each element of the method operates, from the synthetic frequency memory to the binary mask memory and the inverse frequency transform. It helps the reader can follow along with their reasoning.
2. This work introduces an innovative parameterization based dataset distillation method, and is the first work of studying from a frequency perspective. It lays a foundation for further research and discussion in this frequency-oriented parameterization methods.

**Weaknesses:**

1. While this work demonstrates the performance increased relative to several previous methods, it did not compare with another parameterization method [1], where the performance seems to be better than this work. It would be good to include this comparison and discuss the potential limitations of the frequency-based parameterization approach.
2. it's unclear how the choice of transform (Discrete Cosine Transform (DCT), Discrete Fourier Transform (DFT), or Discrete Wavelet Transform (DWT)) might influence the performance or effectiveness of their approach. This aspect is not thoroughly investigated or discussed.
3. The Spectral Bias could impact the network's ability to generalize to new data, there is little discussion on this which could hurt the effectiveness of the frequency domain-based approach presented in the paper.

[1] Deng, Zhiwei, and Olga Russakovsky. "Remember the past: Distilling datasets into addressable memories for neural networks." Advances in Neural Information Processing Systems 35 (2022): 34391-34404.

**Questions:**

1. In section 3.3, can the assumption that the function F is linearly bijective limit the applicability of the method? Is there any situation where this assumption might not hold and how would that affect the results?
2. How does the approach handle high-dimensional datasets? Is there a limit on the dimensionality that the method can effectively handle?

**Limitations:**

The authors did not include the limitation section, it will be good to have a discussion on cross architecture generalization ability of the proposed method.

---

> ### Author Rebuttal · Authors · 2023-08-09
>
> We really appreciate your constructive review on our manuscript.
>
> $\textbf{[W1. Comparison with [1]]}$
>
> Thank you for your valuable suggestion. [1] introduces a new parameterization method called memory addressing (MA). The key concept is to create a common representation by encapsulating the features shared among different classes into a set of bases. As the reviewer noted, the reported performances of [1] show mixed results under various settings. However, the performance of [1] is not solely due to the MA but also includes the effects of other components. For a fair comparison between MA and FreD, we standardized the distillation loss and evaluated their performances.
>
> Table 1 in the rebuttal material shows that FreD and MA exhibit competitive performances with each other under the implemented settings of DM and TM losses. In the same Table,  we further assessed the transferability of each approach by evaluating the robustness of the synthetic datasets against diverse distribution shifts. FreD particularly shows better performances than MA in most corrupted versions of datasets. We conjecture that because FreD selects informative dimensions in the frequency domain, it has good robustness to corruptions that typically occur in the high-frequency domain. Please refer to the response to Reviewer Xfn3. In terms of computational cost, MA requires roughly twice the computation time of FreD. We will include further comparisons with MA in the revised version.
>
> $\textbf{[W2. The impact of utilizing different frequency transforms on FreD]}$
>
> Please refer to Q1 in the global response.
>
> $\textbf{[W3. Discussion on Spectral bias and the influence on the frequency domain-based approach]}$
>
> We appreciate the reviewer's comment. Spectral bias refers to the phenomenon that neural networks are prone to prioritize learning the low-frequency components over relatively higher-frequency components [2,3,4]. For FreD, even though we employed a mask based on the explained variance ratio (EVR), which is not a simple low-frequency filter: the low-frequency components in the frequency domain were predominantly selected in most experimental scenarios. It should be noted that our EVR does not enforce to keep the low-frequency components, unlike the neural network's spectral bias; what EVR only enforces is keeping the components with higher explanation ratio on the image feature.
>
> However, this characteristic can become a risk to performance if the task-specific information of data is mostly found in the high-frequency components. These cases include 1) medical imaging on fine details like tumors [5] and 2) digital watermarking [6]. In such cases, there may be a requirement for a new masking that allows FreD to capture important high-frequency components. The masking strategy of FreD can be flexibly operated, and depending on the characteristics of the given dataset and task, it can readily employ other strategies as needed.
>
> $\textbf{[Q1. The impact of linear bijectivity of $\mathcal{F}$ on the applicability and performance of FreD]}$
>
> $\textbf{Impact of linear bijectivity}$ As the reviewer has pointed out, for the theory presented in section 3.3 to hold, the function $\mathcal{F}$ must be linearly bijective. Note that FreD could utilize any kind of frequency transform, such as DCT, DFT, and DWT. Since these transforms are all linearly bijective in theory, the proposed theorems could be applicable without any limitation.
>
> Having said that, we elaborate the potential issues that might emerge, when integrating not linearly bijective transforms into the framework of FreD, as follows:
>
> $\textbf{When $\mathcal{F}$ is not bijective}$ If $\mathcal{F}$ is not bijective, then its inverse $\mathcal{F}^{-1}$ does not exist. This absence makes the process of transforming to another domain and then restoring back to the original domain infeasible. It potentially results in information loss that interferes accurate reconstruction of the original image. An alternative solution could be separating the transform into distinct encoder and decoder components, which enable a procedure for one-to-one mapping. However, the construction of such components necessitates additional training costs, when $\mathcal{F}$ of FreD does not require any kind of additional training. Furthermore, if the encoding is not one-to-one, information might be lost during the encoding process.
>
> $\textbf{When $\mathcal{F}$ is not linear but bijective}$ FreD employs a subset of dimensions in the frequency domain. The choice is feasible as the linearly bijective transform maintains the EVR in the selected dimensions. Proposition and theorem in section 3.3 theoretically support this attribute, where these do not apply to nonlinear bijective transforms. For 2D-image processing, nonlinear bijective transforms include 1) Log-Polar Transform, and 2) Radial Basis Function (RBF) transform. Domains from these transforms do not exhibit the concentration of the original dataset's variance on specific dimensions.
>
> $\textbf{[Q2. Applicability and Efficacy of FreD on high dimensional datasets]}$
>
> Please refer to Q2 in the global response.
>
> $\textbf{[L1. Limitation of Frequency domain-based approach]}$
>
> Please refer to Q3 in the global response.
>
> $\textbf{[L2. Cross Architecture Generalization of FreD]}$
>
> Please refer to Table 2 in the main paper for the performance of cross architecture generalization. FreD overwhelms other baselines over the whole setting.
>
>
> [1] Remember the Past: Distilling Datasets into Addressable Memories for Neural Networks, NeurIPS2022
>
> [2] On the spectral bias of neural networks, ICML2019
>
> [3] Overview frequency principle/spectral bias in deep learning, arXiv2022
>
> [4] Watch your up-convolution: Cnn based generative deep neural networks are failing to reproduce spectral distributions, CVPR2020
>
> [5] MRI based medical image analysis: Survey on brain tumor grade classification, BSPC2018
>
> [6] Digital watermarking for deep neural networks, IJMIR2018

---

> > ### Comment · Reviewer_ajn2 · 2023-08-20
> >
> > Thank you for answering my concerns, however, regarding the first point, did you try with FreD + Bptt backbone? Bptt (with momentum) is another standard optimization framework for dataset distillation and it would be essential to evaluate FreD's efficiency with it.

---

> > > ### Author Response · Authors · 2023-08-20
> > > **Response for reviewer ajn2**
> > >
> > > Thank you for your valuable response.
> > >
> > > As the reviewer noted, back-propagation through time (BPTT) is another optimization framework that effectively solves the bi-level optimization problem of dataset distillation. FreD is a new type of parameterization framework for dataset distillation, while BPTT is introduced as a new optimization framework for dataset distillation. Hence, they can be utilized orthogonally. To verify the efficiency of FreD, we conduct an experiment on models that combine the BPTT framework with FreD.
> > >
> > > The table below shows the performance of the model with FreD in the BPTT framework on IPC=1 for CIFAR-10. As mentioned in [1], we considered two variants of BPTT framework with and without augmentation. We reduced the number of training iterations for each baseline and FreD from 50000 to 5000 due to the time constraint imposed by the discussion period.
> > >
> > > |Model|Accuracy (%)|
> > > |:---|:---:|
> > > |BPTT                                                      |46.3 &plusmn; 0.7|
> > > |BPTT (reported, Iteration=50000)          |49.1 &plusmn; 0.6|
> > > |**BPTT w/ FreD**                                   |**57.4 &plusmn; 0.4**|
> > > |BPTT + Aug                                            |45.9 &plusmn; 0.5|
> > > |**BPTT + Aug w/ FreD**                         |**58.9 &plusmn; 0.5**|
> > > |**TM w/ FreD** (reported in our main paper) |**60.6 &plusmn; 0.8**|
> > >
> > > As a result, BPTT with FreD outperformed BPTT without FreD under BPTT framework regardless of whether or not the augmentation was used. Furthermore, even when compared to the performance of BPTT with full iteration training reported in the original paper (49.1 &plusmn; 0.6), BPTT w/ FreD achieved higher performance (57.4 &plusmn; 0.4). It indicates that FreD is an efficient methodology that can also be applied in the BPTT framework. Furthermore, TM w/ FreD, a different optimization framework that we basically used in the main paper, performed 60.6 &plusmn; 0.8 on IPC=1 for CIFAR-10. We want to note that TM w/FreD outperforms basic BPTT by about 14%p. We will add these experimental results in the revised version.
> > >
> > > [1] Remember the Past: Distilling Datasets into Addressable Memories for Neural Networks, NeurIPS2022

---

### Author Rebuttal · Authors · 2023-08-09

$\textbf{[Q1. The impact of utilizing different frequency transforms on FreD]}$

We utilized three frequency transforms as the reviewers noted: DCT, DFT, and DWT. We especially want to highlight the energy compaction property of DCT, where most of the signal information tends to be concentrated in a few low-frequency components (Please refer to Figure 2 in [1] and Figure 1 in [2]). This characteristic aligns well with the motivation of FreD, and Figure 7-(b) in the main paper demonstrates that DCT is the best choice among the possible options.

To further analyze the effect of frequency transforms on FreD, we have conducted various experiments. Table 3 in the rebuttal material presents the results as follows:

* Across all settings, we observe improved performances of FreD than the baseline regardless of the type of frequency transform employed.

* DCT outperforms DFT and DWT in most cases, highlighting the effective exploitation of DCT's energy compaction property within the FreD framework.

* DFT exhibits relatively lower performance in comparison to DCT and DWT. This discrepancy is attributed to the complex-valued nature of DFT. Unlike DCT and DWT, which operate in real frequency space, DFT requires additional resources to represent a single instance due to its complex space. As a result, the quantity of synthetic instances that can be generated within an identical budget is reduced by half than others, leading to lower performance.

$\textbf{[Q2. Applicability and Efficacy of FreD on High dimensional dataset]}$

In the spatial domain, discerning the informative dimensions is fundamentally challenging. As a result, dataset distillation must consider all dimensions within the spatial domain during the optimization. If all dimensions are not taken into account, it inevitably leads to a loss of information. In contrast, FreD selectively optimizes the essential dimensions by utilizing the transform to the frequency domain, thus it potentially magnifies its efficiency as the data dimension increases. We conjecture that this property is the main reason for the improved performance of FreD than other baselines.

To support our conjecture, we report the performances of FreD and other methods on Tiny-ImageNet ($3\times64\times64$) and ImageNet-Subset ($3\times128\times128$) in Table 1,2 in the supplementary material. From the results, FreD outperforms the baselines by a large margin in whole settings. Additionally, in the rebuttal material, we also report the performance on LSUN dataset ($3\times128\times128$) and additional results on ImageNet-Subset in Figure 4 and Table 4, respectively. FreD empirically shows consistently improved performances than the baselines. These results demonstrate the effectiveness of frequency domain-based perspective in dataset distillation.

$\textbf{[Q3. Limitation of FreD]}$

$\textbf{1. Efficacy differences depending on the applied domain}$ The applicability and efficacy of FreD's frequency transform are demonstrated specifically within the spatial domain of 2D/3D images. Among the data domains commonly used in machine learning framework, natural language domain would likely be challenging to connect with the frequency domain directly. While Fourier Transform can still be applied to text after it's been converted into a numerical format, such as a time series, this conversion is non-trivial and the resulting frequency domain representation may not be as intuitively meaningful. Thus, for certain domains, the effectiveness of frequency transform may not be as substantial as it is for 2D/3D images.

However, in multi-modal tasks, major components like audio signals and video data naturally align with the frequency domain. Tools such as the 1D Fourier transform for audio signals and 3D Fourier transform for video data already exist to process these types directly. Excluding a few specific domains, FreD would be a framework that can be applied across a broader range of domains.

$\textbf{2. Performances highly dependent on masking strategy}$ FreD's frequency-based parameterization is motivated by the fact that the spatial domain information of a 2D image can be concentrated in specific components of the transformed frequency domain. The explained variance ratio (EVR) based masking selects important dimensions from the frequency domain, consistently showing strong performances across the various experiments conducted in this study.

Having said that, results in Figure 7-(a) of the main paper demonstrate that there could be significant performance disparities depending on the masking strategy employed. Therefore, there could be substantial issues if the chosen masking strategy fails to select the important dimensions accurately. For certain datasets or tasks, essential task-specific information might be contained in the high-frequency region. These cases could include 1) medical imaging on fine details like tumors [3] and 2) digital watermarking [4]. This highlights a limitation that EVR masking may not be suitable for all data and tasks.

It should be noted that the masking strategy of FreD can be flexibly operated, and depending on the characteristics of the given dataset and task, it is not restricted to using an EVR-based mask and can readily employ other strategies as needed.

One potential solution to identify task-related frequency components is to utilize the gradient of the given task loss. If specific frequency components have a substantial gradient distribution, it indicates that the component greatly influences the task. By substituting with gradient-based masking, we could address the potential limitations that EVR mask might have.

[1] Discrete Cosine Transform, IEEE Trans. Comput. 1974

[2] Compression, Restoration, Resampling, “Compressive Sensing”: Fast Transforms in Digital Imaging, arxiv2015

[3] MRI based medical image analysis: Survey on brain tumor grade classification, BSPC2018

[4] Digital watermarking for deep neural networks, IJMIR2018

---

### Author Response · Authors · 2023-08-16
**Thank you for the reviews. Any comments for further clarification?**

Dear Reviewers,

We truly appreciate your constructive comments on our work and thank you so much for your time.

Based on the valuable reviews, we have tried to clarify how our proposed idea would work, its applicability, and its efficacy.
Also, we have attached the rebuttal material which contains additional experimental results to support our explanations.

We are willing to provide further explanation on any unclear point.
Please let us know if there's anything we should have expressed more clearly.

Best regards.

---

### Comment · Area_Chair_3esP · 2023-08-18
**Please look at the authors' reply**

Dear Reviewers,

Please do look at the authors' rebuttal if you have not done so. Please let the authors know if they have addressed your concerns.

Thanks for your contribution to NeurIPS.

AC

---

### Decision · Program_Chairs · 2023-09-21

**Decision:**

Accept (poster)

**Comment:**

The manuscript has been reviewed by four reviewers, where the ratings are respectively 4,5,6,8.

The 4-rating review is the only one on the negative side. The only remaining concern from this reviewer is the FreD + Bptt.

The authors have demonstrated good performance on this setting in the rebuttal (despite the review didn't reply to this last post).

As such, it is considered all the reviewers all in general satisfied with the manuscript. The AC agrees with this consensus and finds no basis to overturn the convergence.

Congrats!